# Genome-resolved metatranscriptomics reveals conserved root colonization determinants in a synthetic microbiota

Nathan Vannier[1,3], Fantin Mesny [1,4,5], Felix Getzke[1,5], Guillaume Chesneau[1], Laura Dethier[1], Jana Ordon [1], Thorsten Thiergart[1] & Stéphane Hacquard [1,2] ✉

The identification of processes activated by specific microbes during microbiota colonization of plant roots has been hampered by technical constraints in metatranscriptomics. These include lack of reference genomes, high representation of host or microbial rRNA sequences in datasets, or difficulty to experimentally validate gene functions. Here, we recolonized germ-free *Arabidopsis thaliana* with a synthetic, yet representative root microbiota comprising 106 genome-sequenced bacterial and fungal isolates. We used multi-kingdom rRNA depletion, deep RNA-sequencing and read mapping against reference microbial genomes to analyse the in planta metatranscriptome of abundant colonizers. We identified over 3,000 microbial genes that were differentially regulated at the soil-root interface. Translation and energy production processes were consistently activated in planta, and their induction correlated with bacterial strains' abundance in roots. Finally, we used targeted mutagenesis to show that several genes consistently induced by multiple bacteria are required for root colonization in one of the abundant bacterial strains (a genetically tractable *Rhodanobacter*). Our results indicate that microbiota members activate strain-specific processes but also common gene sets to colonize plant roots.

As photoautotrophs, plants assimilate carbon into complex organic compounds and invest a substantial amount of these photoassimilates in the rhizosphere, thereby sustaining microbial growth[1]. Consequently, they host a large diversity of microbes within and on their roots that include primarily bacteria and fungi[2–4]. Microbiota establishment in roots is governed by a complex interplay between the microbiota, the host, and the environment[5–12], as well as by microbe–microbe interactions[13–16].

Because root-associated microbes have been interacting with host-derived cues for ~450 million years, they have likely evolved complex mechanisms to colonize, survive and thrive in the root environment[17,18]. While the mechanisms by which model mutualistic

and pathogenic microorganisms colonize root tissues have been intensively studied[19–21], our knowledge of the processes activated by phylogenetically diverse root commensals to colonize and persist in roots within a microbial community remains fragmented. Bacterial mechanisms of root colonization were recently identified in individual root isolates using multiple approaches, including experimental evolution[22], comparative genomics[23], randomly-barcoded transposon mutagenesis sequencing[24,25] (RB-TnSeq) or transposon insertion sequencing[26] (IN-Seq). These bacterial genetic determinants required for root/rhizosphere colonization are primarily involved in chemotaxis towards the roots, primary and secondary attachment to the root surface, and to root hairs[27]. However, it remains unclear whether these

[1]Department of Plant Microbe Interactions, Max Planck Institute for Plant Breeding Research, 50829 Cologne, Germany. [2]Cluster of Excellence on Plant Sciences, Max Planck Institute for Plant Breeding Research, 50829 Cologne, Germany. [3]Present address: IGEPP, INRAE, Institut Agro, Univ Rennes, 35653 Le Rheu, France. [4]Present address: Institute for Plant Sciences, University of Cologne, 50923 Cologne, Germany. [5]These authors contributed equally: Fantin Mesny, Felix Getzke. ✉e-mail: hacquard@mpipz.mpg.de

processes are activated in a community context during root colonization and more importantly, whether diverse root colonizers deploy similar or dissimilar strategies to colonize plant roots.

Development of metatranscriptomics-based approaches were successfully used to identify microbial functions expressed ex planta in rhizosphere/rhizoplane habitats under different ecological contexts[28–32]. However, bacterial transcriptome profiling remains more challenging in planta and has been hampered by numerous technical constraints, already in mono-association experiments (see recent examples for leaf commensals and pathogens)[33–35]. These include (i) the dominance of plant-derived rRNA and mRNA and (ii) the complexity and diversity of the root microbiota that limit transcriptome coverage of individual strains and prevent unambiguous read mapping between closely related species. These constraints, together with difficulties linked to genome annotations from complex samples, have hampered the investigation of microbial functions and molecular mechanisms concomitantly deployed by multiple strains during root microbiota establishment.

The recent establishment of comprehensive culture collections of root-associated bacteria and fungi that are representative of naturally occurring root microbiota[13,36,37] offers the opportunity to design multi-kingdom synthetic communities (SynComs) and to reconstitute the root microbiota of healthy plants under strictly controlled environments[13]. The use of genome-sequenced microbial SynComs has two main advantages compared to natural samples regarding metatranscriptome profiling: (i) the control over the community composition allows building taxonomically diverse communities with reduced complexity thus yielding higher average transcriptome coverage and (ii) the genomes of the cultured organisms are used as reference for read mapping. Reference-based metatranscriptomics has multiple advantages compared to metatranscriptome assembly including better detection of genes (as transcripts with a low number of reads cannot be assembled), a more accurate read assignment to the different microbiota members and consequently an overall higher number of reads mapped. We hypothesized that this strategy could enable the simultaneous inspection of multi-kingdom transcriptomes (bacteria, fungi, and the host plant) at strain-level resolution in a community context.

Here, we repopulated roots of germ-free *Arabidopsis thaliana* with a multi-kingdom microbial SynCom composed of 84 bacteria and 22 fungi in the gnotobiotic FlowPot system[38] and profiled changes in both community structure and the transcriptomes of bacteria and fungi at the soil–root interface. We analyzed microbial transcriptional reprogramming at both strain- and community-level resolution and identified microbial processes induced during root colonization. We experimentally validated that several bacterial genes induced upon host contact by multiple strains contribute to bacterial proliferation at roots. This study identifies conserved and strain-specific processes required for microbial adaptation to the root environment in a community context.

## Results

### High-resolution read mapping against host and SynCom reference genomes

We tested whether multi-kingdom ribosomal RNA (rRNA) depletion, combined with deep RNA sequencing and subsequent reference genome-guided read mapping allowed to capture the host and SynCom metatranscriptome. Simultaneous depletion of plant, fungal, and bacterial-derived rRNA was successfully achieved, resulting in >25-fold enrichment of non-rRNA over rRNA sequence reads for both root and matrix samples (Supplementary Fig. 1a, b; increase from <5% to >75% in both root and matrix samples). Illumina-based sequencing of these rRNA-depleted samples (matrix: 37–44 million reads; roots: 64–158 million reads), followed by pseudo-mapping of trimmed reads against the 825,409 coding sequences derived from *A. thaliana* as well as the

84 bacterial and 22 fungal genomes revealed that 41.9% (15–16 million reads) could be mapped to microbial genomes for matrix samples and 8.7% (4–16 million reads) for root samples (Supplementary Fig. 1c, "Methods"). As expected, a large proportion of reads mapped to the *A. thaliana* reference genome for root (51%) but not matrix (0.2%) samples, whereas several unassigned or rRNA reads remained unmapped for both root (40.4%) and matrix (57.9%) samples (Supplementary Fig. 1c). After gene duplicate removal (see method and Fig. 1a, b), 73 million reads could be mapped to 73 out of 84 bacterial genomes across all samples on a total of 72,984 unique bacterial genes (genes detected per strain: min = 1 gene, max = 5242 genes, average = 1376 genes, Fig. 1a) and 1,657,895 reads could be mapped to the 22 fungal genomes on a total of 21,704 unique fungal genes (min = 29 genes, max = 4504 genes, average = 986 genes, Fig. 1b). In contrast to fungi, we observed high transcriptome coverage for several bacteria for which >50% of the genes could be detected as expressed (matrix: 15 bacterial strains, root: 12 bacterial strains, Fig. 1a). In root samples, we detected 20,265 *A. thaliana* genes as expressed and observed that most of these genes (i.e., 19,802, 97.7%) overlapped and showed consistent expression levels with those of a previous study reporting *A. thaliana*'s response to a multi-kingdom synthetic community (183 bacteria, 24 fungi, 7 oomycetes) in the same gnotobiotic plant system[9] (Pearson, $r = 0.75$; $P < 0.001$, Supplementary Fig. 1d). We conclude that multi-kingdom rRNA depletion, deep RNA sequencing and reference-based read mapping represents a suitable strategy to simultaneously capture the transcriptome of the host and associated abundant bacterial and fungal strains, but remains insufficient to capture the entire SynCom metatranscriptome.

### RNA-Seq read mapping reveals SynCom structure at the root interface

We asked whether our RNA-Seq read mapping approach against SynCom reference genomes can be used to profile SynCom diversity and composition at the root interface. We co-purified DNA from the same samples used for RNA-Seq and performed amplicon sequencing using primers targeting the bacterial 16 S rRNA gene and the fungal ITS (see "Methods"). Comparison between RNA- and DNA-based profiling revealed high overlap in the number of detected strains, with 46% of the bacteria (39 out of 84 for matrix) and 90% of the fungi (20 out of 22 for matrix) consistently detected by both methods in matrix samples and 46% and 87%, respectively, in roots samples (Fig. 2a, b). Notably, RNA-Seq read mapping detected more strains than the amplicon-based method (Matrix/Roots: 80%/76% vs. 49%/48% for bacteria; 100%/100% vs. 90%/86% for fungi, Fig. 2a, b). In a second step, we aggregated bacterial and fungal read counts from both DNA and RNA datasets at class-, order-, family-, genus-, or species-level resolution. This revealed that RA profiles were largely consistent, irrespective of the method or the taxonomic resolution (Supplementary Fig. 2). The bacterial community was dominated by Xanthomonadales, Actinomycetales, Pseudomonadales, and Rhizobiales, whereas Hypocreales and Glomerallales prevailed in the fungal community (Fig. 2c, d). Enrichment tests conducted for these abundant groups indicated that Pseudomonadales RA was significantly reduced in roots vs. matrix for both DNA- (DESeq2, $P = 0.018$, log2 fold change (FC) = −2.51) and RNA-based (DESeq2, $P < 0.001$, log2FC = −3.02) approaches, whereas a significant change in the RA of Rhizobiales (DESeq2, $P = 0.028$, log2FC = 0.82) and Hypocreales (DESeq2, $P < 0.001$, log2FC = −2.19) was only detected by RNA-based profiling (Supplementary Data 1). RNA-based abundance profiling revealed shifts in relative abundance (RA) profiles at strain-level resolution between matrix and root samples (Fig. 1a, b), with eight bacterial and four fungal strains significantly enriched in roots vs. matrix and seven bacteria and a single fungal strain showing the opposite pattern (Supplementary Fig. 3). Notably, RAs of microbial strains that were consistently detected by both approaches (see overlap in Fig. 2a, b) were significantly correlated between RNA- and

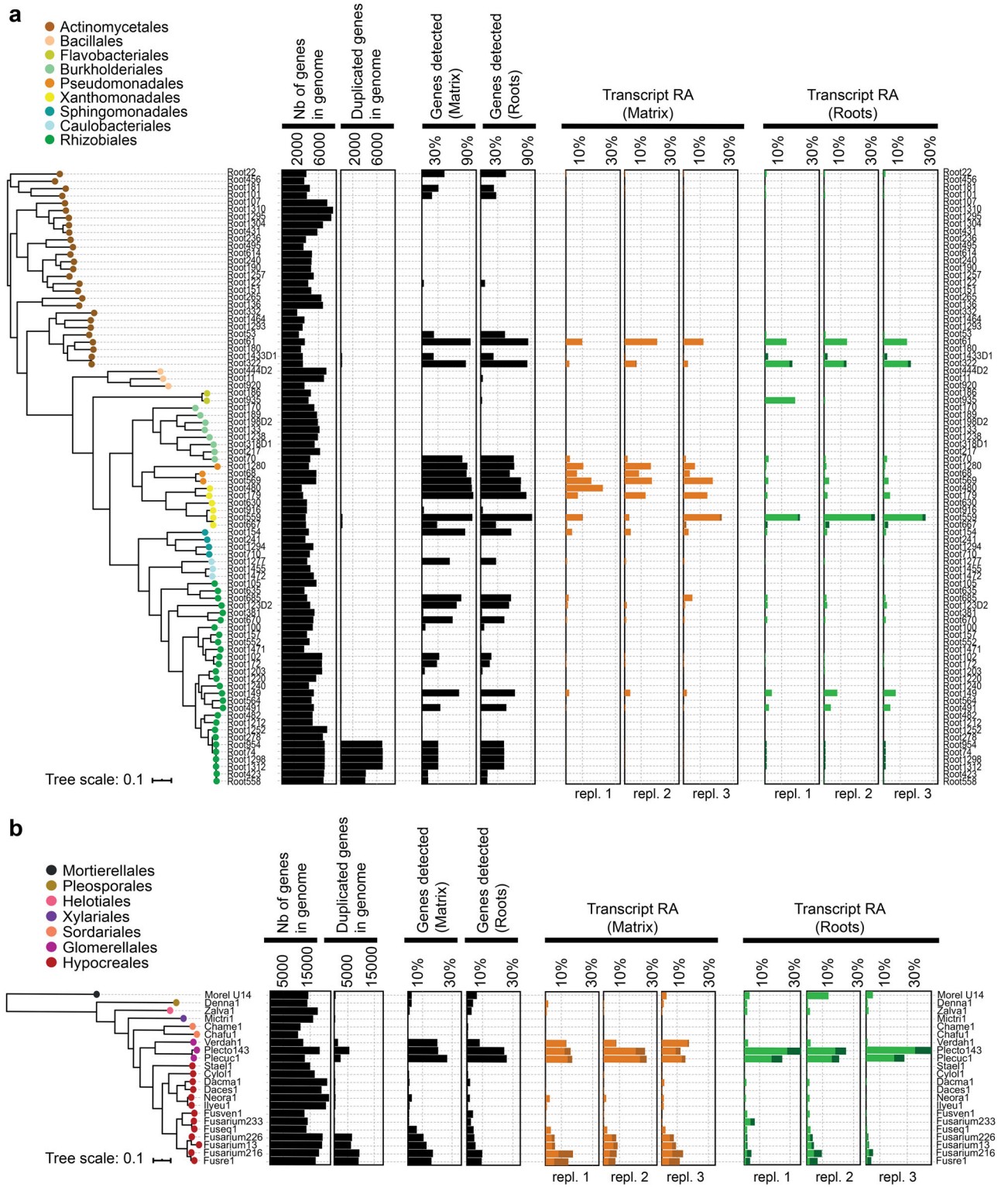

**Fig. 1 | RNA-based detection of SynCom members in the gnotobiotic FlowPot system. a, b** Whole-genome phylogenetic tree of bacterial ($N = 84$) (**a**) and fungal ($N = 22$) (**b**) strains. The number of predicted genes in genomes and the number of duplicated genes between strains are shown next to the trees (black bars). The next two columns indicate the average percentage of genes for which at least one RNA- seq read mapped the corresponding reference genome in at least one matrix or roots sample (black bars, $n = 3$). Relative abundance (RA) of individual strains based on RNA-Seq read mapping is shown for each of the three replicates. Matrix: orange. Roots: green. Transcripts mapped to duplicated genes are shown in dark orange (matrix) and dark green (roots). Source data are provided as a Source Data file.

DNA-based profiling for both matrix and root samples (Fig. 2e, f). We conclude that reference genome-guided RNA-Seq read mapping can be used to understand the functional but also structural architectures of multi-kingdom microbial consortia.

## Extensive transcriptional reprogramming during root colonization at a strain-level resolution

We performed differential expression analyses for individual strains (DeSeq2[39]) to identify microbial genes induced upon host contact. We

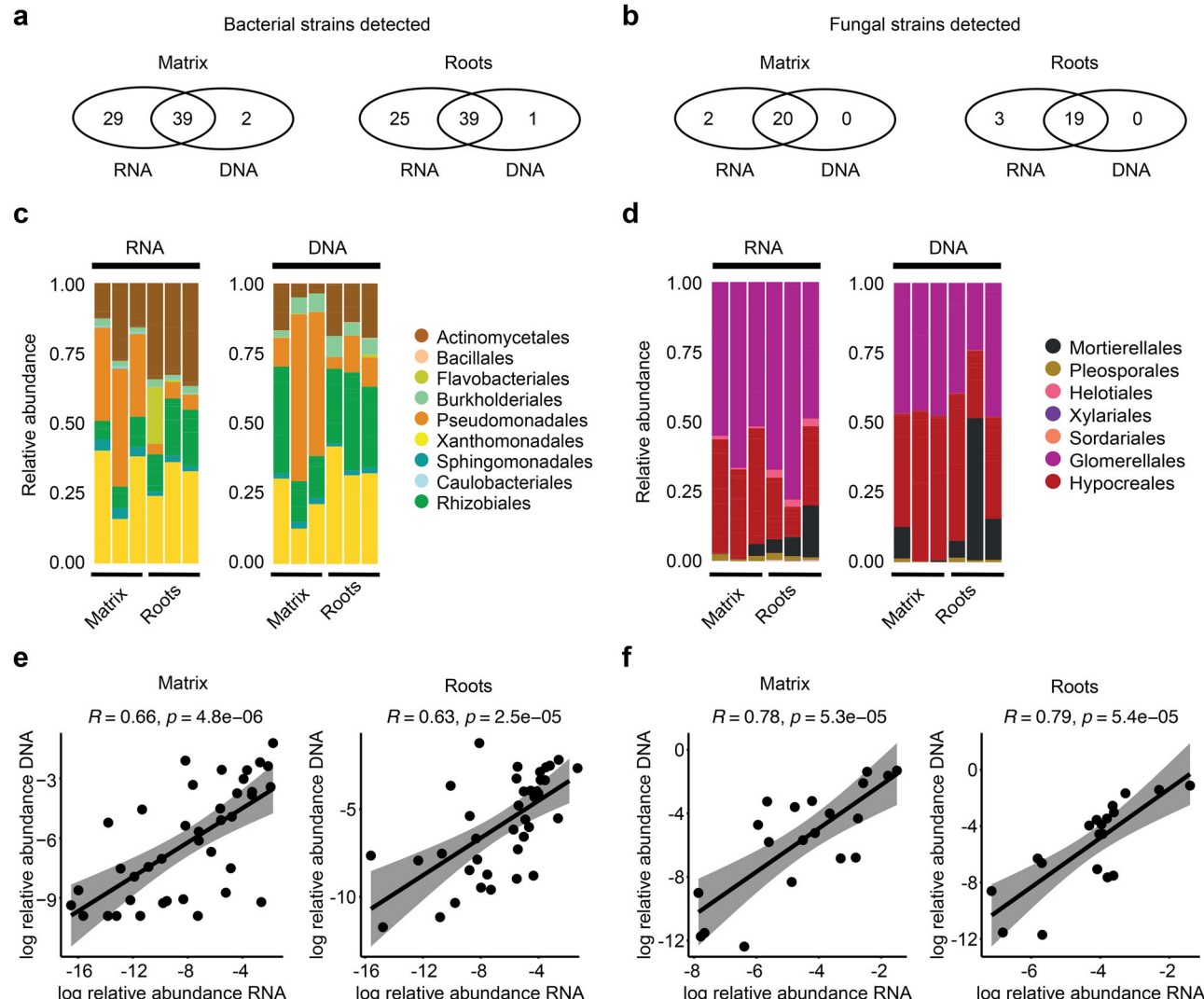

**Fig. 2 | Comparison between RNA-based and DNA-based profiling of SynCom members. a, b** Bacterial (**a**) and fungal (**b**) strains detected in matrix and root samples by RNA-Seq read mapping (i.e., at least one transcript mapped to a strain reference genome in any of the replicates, RNA, *n* = 3) and/or by DNA-based 16 S rRNA (**a**) and ITS (**b**) amplicon sequencing (DNA, *n* = 3). Note that 84 bacteria and 22 fungi were used in the initial inoculum. **c, d** Relative abundance of bacterial (**c**) and fungal (**d**) strains aggregated at the order level (*y* axis) in roots and matrix samples using RNA-based and DNA-based profiling. **e, f** Relative abundance (mean of three replicates, log-transformed) of bacterial (**e**) and fungal (**f**) strains in the matrix and root samples using RNA-based and DNA-based profiling. Bacterial (**e**) and fungal (**f**) strains detected by both approaches (see **a, b**) were used for correlation analyses (bacteria: *N* = 39 for matrix, *N* = 39 for roots; fungi: *N* = 20 for matrix, *N* = 19 for roots). *P* values and correlation scores obtained from a Pearson correlation test are shown above the graphs. Gray area represents the 0.95 confidence interval of the linear correlations. Source data are provided as a Source Data file.

identified 3068 differentially expressed genes (DEGs) between root and matrix samples for bacteria (2007 up and 1061 down, Supplementary Data 2) and 94 DEGs for fungi (50 up and 44 down, Supplementary Data 3), which is consistent with the low RNA-Seq read coverage observed for these filamentous eukaryotes (Fig. 1b) and earlier observation showing that bacterial commensals strongly inhibit fungal growth and prevent fungal dysbiosis in roots[12,13]. Fungal DEGs derived from core mycobiota members[40] did not display specific functional enrichment (topGO[41], *P* > 0.05). Several encode secreted proteins (18/94; SignalP3)[42], candidate effectors (4/94, SignalP3+EffectorP4)[42,43], carbohydrate-active enzymes (8/94, dbCan5)[44] or were previously shown to mediate host–fungi associations (11/94, PHIBase6)[45]. The topmost upregulated genes in planta encode secreted effectors, cerato-platanin-secreted proteins, peptidases-like subtilisin and proteases, or homologs of previously described virulence factors that might have relevance for host colonization[46–49] (Supplementary Data 3).

For bacteria, we observed that DEGs of strains that were abundant at roots (i.e., 20 strains, >1000 genes detected in both roots and matrix samples, Fig. 3a) encompassed ~95% of all bacterial DEGs (2899 DEGs out of 3068, Supplementary Data 2). We validated that read counts per gene were normally distributed for these strains and that housekeeping genes showed the expected absence of differential regulation (Supplementary Fig. 4 and "Methods"). Given that root samples contain on average four times less microbial reads than matrix samples (i.e., high representation of plant-derived reads, See Supplementary Fig. 1), DESeq2[39] differential expression testing was less confident at identifying down- than upregulated genes between roots and matrix for several strains (Fig. 3a and Supplementary Fig. 5). Inspection of the topmost induced genes in planta (log2FC > 11, Table 1) revealed functional involvement in fimbriae/pili formation (Root1280|1676, Root154|1225)[27], synthesis of glucosamine-6-phosphate (Root1277|481: *glmS*), cell wall remodeling (Root154|2017), iron uptake (Root154|382: *tonB*), antibiotic efflux

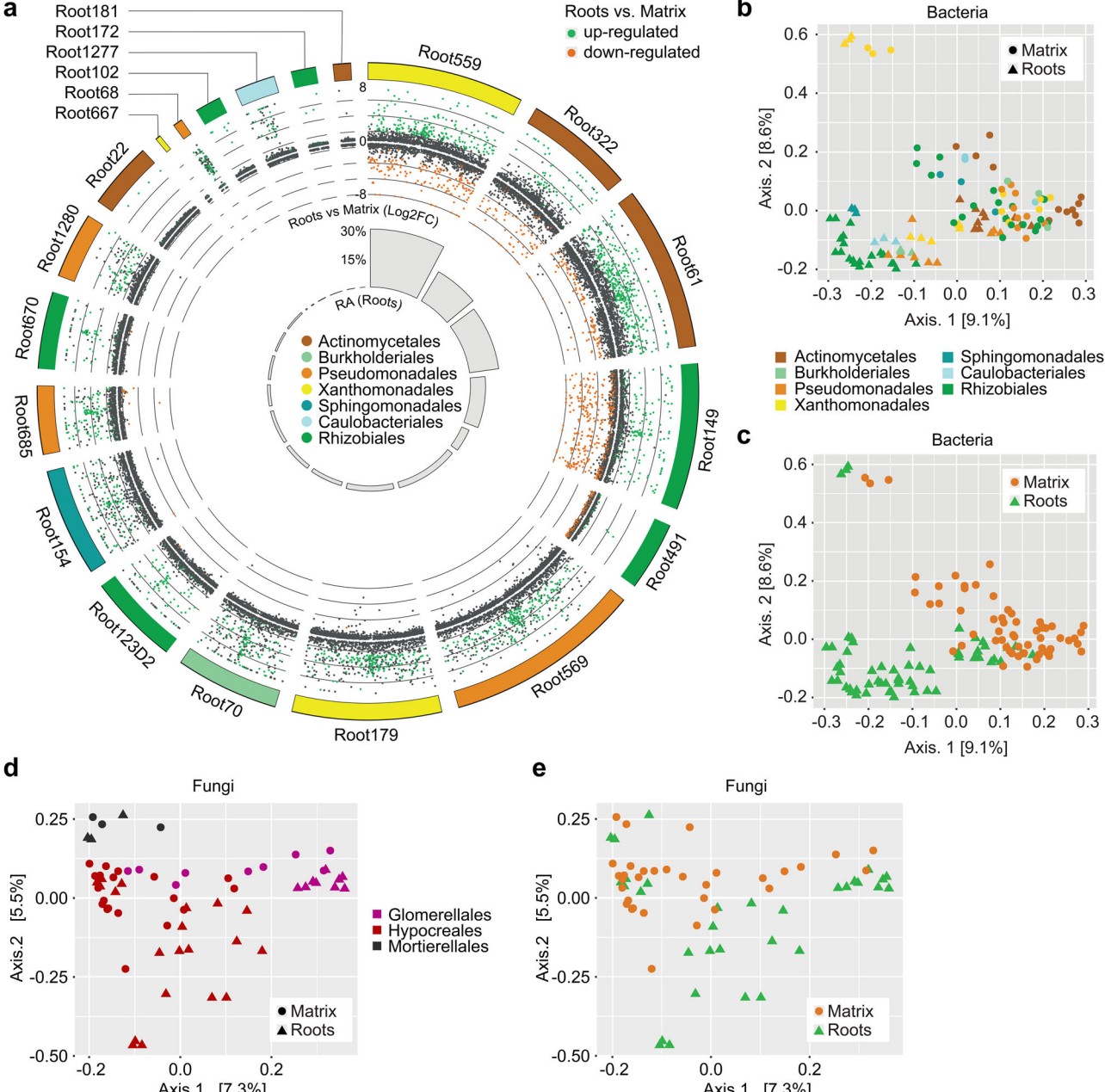

**Fig. 3 | Differential gene expression between roots and peat matrix for robust bacterial root colonizers. a** Circle plot (computed with circos[83]) depicting differentially expressed genes (DEGs) for the top 20 abundant bacterial root colonizers. DEGs (roots vs matrix samples) were computed independently for each strain using DESeq2[39]. The outer ring indicates strain phylogeny (order), transcripts log2FC is represented on the *y* axis (8, 0, −8 log2FC scale) and transcripts are colored in green if upregulated in roots (log2FC > 1.5), orange if downregulated in roots (log2FC < −1.5) and dark gray if not differentially enriched between roots and matrix. The inner gray barplot depicts the relative abundance (%) of each strain in

root samples (see also Fig. 1a). Principal coordinate analysis based on Bray–Curtis distances between bacterial strains in all samples computed on orthogroup relative abundances (relative number of transcripts) colored by orders (**b**) and sample types (**c**). All samples are represented for each strain (i.e., 6 samples per strain). Principal coordinate analysis based on Bray–Curtis distances between fungal strains in all samples computed on orthogroup relative abundances (relative number of transcripts) colored by orders (**d**) and sample types (**e**). All samples are represented for each strain (i.e., 6 samples per strain). Source data are provided as a Source Data file.

(Root70 | 3816: *llpE*), bacterial pathogenesis (Root154 | 1686: *rhlE2*) or formate production (Root70 | 4176: *purU*), as well as two genes with unknown functions (Root154 | 1230, Root154 | 768). To further validate that our approach identified bacterial sets of genes that are specific to host colonization, we performed a second reconstitution experiment with the same 84-member bacterial SynCom inoculated in the presence or absence of the 22-member fungal SynCom in matrix samples in the absence of the host (see "Methods"). We identified only 191 bacterial DEGs that responded to the fungal SynCom

(Matrix+F vs. Matrix-F in the absence of the host, Supplementary Fig. 6a), contrasting with the 3068 bacterial DEGs that responded to the presence of the host (Roots vs. Matrix in the presence of fungi, Supplementary Data 2). No overlap was observed between these two sets of DEGs (0.15%), whereas extensive overlap did exist between the sets of expressed genes (56.44%) (Supplementary Fig. 6b). We conclude that extensive and host-dependent transcriptional reprogramming occurs in multiple bacterial species during root microbiota establishment.

**Table 1 | Top 50 most highly induced bacterial genes in roots vs. matrix**

| Strain ID\|Gene ID | Gene name | Base mean[a] | log2FC[b] | Adjusted P[c] | Function[d] |
|---|---|---|---|---|---|
| Root1280\|1676 | | 645 | 16.8 | 1.31e-16 | Type-1 fimbrial protein, A |
| Root154\|1230 | | 857 | 16.4 | 6e-13 | Unknown |
| Root1277\|481 | glmS | 1157 | 16.3 | 4.21e-06 | Glutamine-fructose-6-phosphate amidotransferase |
| Root154\|382 | | 838 | 16.1 | 6.35e-30 | TonB-dependent receptor |
| Root70\|3816 | llpE | 634 | 15.2 | 6.54e-25 | Alpha-beta hydrolase fold-3 domain protein |
| Root154\|1686 | rhlE2 | 326 | 14.6 | 8.7e-08 | DEAD DEAH box helicase |
| Root154\|2017 | | 1424 | 14.3 | 6.9e-10 | Mycolic acid cyclopropane synthetase |
| Root154\|1225 | | 1096 | 14.1 | 2.78e-12 | Pili and flagellar-assembly chaperone |
| Root70\|4176 | purU | 345 | 14.1 | 3.01e-22 | Formyltetrahydrofolate deformylase |
| Root154\|768 | | 1251 | 14.1 | 3.56e-08 | Unknown |
| Root1277\|4059 | Echdc | 257 | 13.9 | 5.93e-05 | Enoyl-CoA hydratase/isomerase |
| Root154\|1227 | | 3781 | 13.8 | 4.16e-16 | Unknown |
| Root123D2\|2583 | rskA | 348 | 13.8 | 6.31e-06 | Anti-sigma-K factor |
| Root1277\|897 | | 242 | 13.8 | 7.56e-05 | Predicted periplasmic protein (DUF2271) |
| Root154\|369 | | 181 | 13.6 | 2.12e-06 | Xylose isomerase-like TIM barrel |
| Root70\|1346 | | 215 | 13.4 | 1.77e-05 | Nucleotidyltransferase |
| Root569\|2962 | pstC | 467 | 13.4 | 3.17e-10 | Polypeptide phosphate ABC transporter subunit |
| Root154\|381 | | 2868 | 13.4 | 3.25e-15 | Lactoylglutathione lyase and related lyases |
| Root70\|3634 | | 202 | 13.2 | 2.65e-05 | Drug metabolite transporter superfamily |
| Root1277\|4057 | | 1229 | 13.2 | 7.44e-09 | Amine oxidase flavin-containing A |
| Root154\|985 | filA | 135 | 13.1 | 3.71e-06 | Membrane-bound metallopeptidase |
| Root559\|706 | | 219 | 12.9 | 1.4e-06 | Belongs to the UPF0312 family |
| Root61\|6 | | 166 | 12.8 | 1.09e-06 | Unknown |
| Root685\|2415 | | 608 | 12.8 | 1.54e-07 | Copper chaperone PCu(A)C |
| Root123D2\|3109 | | 192 | 12.6 | 0.0001 | Acetyltransferase (GNAT) domain |
| Root154\|380 | | 91 | 12.6 | 3.3e-06 | PFAM TonB-dependent Receptor |
| Root102\|1333 | | 1413 | 12.6 | 2.35e-05 | Unknown |
| Root102\|4013 | gabD | 841 | 12.5 | 4.26e-05 | Aldehyde dehydrogenase family |
| Root70\|441 | petB | 754 | 12.4 | 2.61e-07 | Cytochrome b6 |
| Root322\|1078 | ilvA | 238 | 12.3 | 1.5e-06 | Pyridoxal-phosphate dependent enzyme |
| Root123D2\|241 | | 674 | 12.3 | 3.66e-07 | Membrane fusion protein (TC 8.A.1) family |
| Root123D2\|1229 | | 154 | 12.2 | 0.0002 | Unknown |
| Root1277\|2958 | dnaA | 403 | 12.1 | 3.58e-05 | Chromosomal replication initiator protein |
| Root172\|4049 | | 559 | 12.1 | 9.61e-05 | von Willebrand factor (vWF) type A domain |
| Root149\|1190 | | 204 | 11.9 | 0.0007 | Uncharacterized protein (DUF2059) |
| Root154\|4234 | rpsP | 53 | 11.6 | 2.24e-05 | Ribosomal protein bS16 family |
| Root322\|1193 | | 155 | 11.5 | 1.93e-05 | Major facilitator Superfamily |
| Root322\|2730 | | 201 | 11.5 | 0.0005 | MFS/sugar transport protein |
| Root149\|1665 | rplQ | 123 | 11.4 | 0.0002 | Ribosomal protein L17 |
| Root670\|456 | dctA | 883 | 11.4 | 3.32e-07 | Aerobic C4-dicarboxylate transporter |
| Root322\|1563 | | 136 | 11.4 | 1.93e-05 | Unknown |
| Root149\|609 | atpF | 115 | 11.3 | 0.0002 | ATP synthase subunit b |
| Root569\|2421 | aspA | 1685 | 11.3 | 1.13e-14 | Aspartate ammonia-lyase |
| Root559\|1423 | rnpA | 90 | 11.3 | 5.94e-05 | Ribonuclease P protein component |
| Root559\|650 | | 99 | 11.2 | 0.0001 | Unknown |
| Root322\|2077 | yigZ | 136 | 11.2 | 0.0001 | Uncharacterized protein family UPF0029 |
| Root61\|2410 | hmuT | 253 | 11.1 | 5.36e-06 | Periplasmic binding protein |
| Root61\|2017 | | 62 | 11.0 | 0.0001 | Domain of unknown function (DUF4407) |
| Root61\|1037 | | 878 | 11.0 | 3.45e-10 | Domain of unknown function (DUF4442) |

[a]DESeq-corrected average number of reads.
[b]Log2 fold change.
[c]DESeq2 Wald test *P* value adjusted for false discovery rate.
[d]Most precise annotation available for the gene.

## A core transcriptional response is activated by multiple strains in roots

To compare genes and functions simultaneously activated by multiple strains in roots, we first predicted orthology between bacterial genes using phylogenetic orthology inference (OrthoFinder[50]). Principal coordinate analysis based on orthogroup (gene family defined by sequence homology; OG) gene expression revealed directional shifts between matrix and root samples that were reproducibly detected across multiple strains for both bacteria and fungi (Fig. 3b–e). PER-MANOVA analyses using distance between strains based on OG expression data revealed that "taxonomy" explained most of the transcriptome differentiation for bacteria, especially at high taxonomic resolution (phylum~6%, class~13%, order 23%, family~40%, species ~55%), followed by "compartment" that explained 6.8% of transcriptome differentiation (Supplementary Data 4). A significant interaction between "compartment" and "taxonomy" was observed, highlighting that a part of the transcriptional reprogramming between roots and matrix is taxon-specific (up to ~15% for the species and 11% for families, Supplementary Data 4). To identify the conserved functions driving transcriptional shifts, we next performed OG-level differential expression analysis at both single strain- and community-level resolution (Fig. 4a). Ranking of OGs according to transcript cumulative log2FC in roots vs. matrix across all strains (cumulative log2FC, Fig. 4a) identified orthologous genes simultaneously activated by multiple bacteria in roots that were also induced at the community scale (community-scale log2FC, Fig. 4a). GO-term analysis revealed that the top-200 OGs with the highest cumulative Log2FC across strains (Supplementary Data 5) were primarily involved in translation (46), unknown functions (26), energy production and conversion (15), replication and repair (15) as well as cell wall remodeling (13) (Fig. 4b). This result was also confirmed by GO-terms enrichments (Fig. 4c and Supplementary Fig. 7) and was consistent with a second independent analysis based on KEGG[51] curated orthology terms (KO) instead of OGs (Supplementary Fig. 8). Most OGs showing the highest cumulated log2FC across strains encode for instance ribosome subunit proteins (*rpsG/J/S, rpoA/B, rplB/C/D/J/M/Q/X*), ATP synthase subunits (*atpA/D/F2/G/H*), a signal recognition particle protein (*ffh*, upregulated by 6 strains), a translational GTPase that regulates virulence in several eubacteria (*typA*, upregulated by 7 strains), a glutamine synthetase (*glnA*, upregulated by 9 strains), or proteins of the phosphate transport system (*pstA/S*, upregulated by 5/8 strains) (Fig. 4a). Upregulation of genes involved in metabolism and transport (inorganic ions, nucleotides, amino acids, carbohydrates and lipids) as well as secretion (i.e., Sec-translocase system proteins *secB/D/E/G*) was also common among the top-200 OGs showing the highest cumulated log2FC in planta, and these terms were also pointed out by a GO analysis (Fig. 4b). The same key bacterial regulons, genes, and functions upregulated in roots were also independently identified by KEGG-based annotation of gene families (Supplementary Fig. 8). Together, our data indicate that a substantial fraction of bacterial processes induced at roots involve evolutionary-conserved functions that promote translation, energy production, and transport, reflecting active bacterial growth and associated high protein biosynthesis rate in the root niche.

## Activation of multiple biological processes associate with bacterial abundance in roots

To test whether the expression of the most upregulated functions correlates with root colonization, we inspected the correlations between above-mentioned GO-term processes activated by bacteria at roots (see 200 OGs shown in Fig. 4a, b) and the RA of the corresponding strains in the root compartment (i.e., RNA-based RA, see Fig. 1a). For each of the 20 selected strains, we used differential expression values (log2FC) obtained for the top-200 OGs (see heatmap in Fig. 4a) and aggregated them by GO terms (Fig. 4b). We then tested whether this average GO-term differential expression correlates with the 20 strains RAs in roots. Because we selected the 200 bacterial OGs showing the highest cumulated log2FC between roots and matrix, induction of several of these functions were highly correlated between each other (Fig. 5a) and likely co-explain the same variation in bacterial RA (see all models in Supplementary Fig. 9). Nonetheless, inspection of the most significant regression model (Fig. 5b) revealed that the average differential expression of the OGs belonging to the GO-term category "Energy production and conversion" (Figs. 4b and 5b) explained 46% of the strain´s RA in the root compartment ($P = 0.0005$, $R^2 = 0.4697$). The results indicate that in planta activation of these processes is strongly linked with bacterial proliferation at roots and therefore suggests a possible link between the ability of diverse bacteria to utilize root-derived cues for their primary metabolism and their ability to dominate in the root microbiota.

## Genes consistently induced by multiple bacteria in roots promote host colonization

To validate the importance of the genes that we identified as upregulated at the soil–root interface, we next focused on the genetically tractable *Rhodanobacter* sp. Root179 (hereafter referred to as Root179), an abundant strain detected in the root compartment (Fig. 3a). We generated a GFP-labeled Root179[48] and confirmed that the strain was able to re-colonize germ-free *A. thaliana* roots in an agar-based plate system (Fig. 6a, see "Methods"). Using a targeted gene knockout approach (see "Methods"), we then deleted Root179 genes of OGs that were consistently upregulated by multiple strains in plant roots, including *typA*, a gene encoding a GTP-binding protein as well as *pstS*, *pstB*, and the full *pstABCS* operon encoding (part of) a phosphate uptake system (see Figs. 4a and 6b and Supplementary Data 2). We also selected three other genes that were specifically and highly upregulated by Root179 during root colonization, including *fimC*–a gene encoding a chaperone protein involved in fimbriae biogenesis, *exbD1*–a gene encoding the biopolymer transport protein *exbD* and *impH-2*–a gene involved in type VI secretion system and biofilm formation (Fig. 6b and Supplementary Data 2). Re-colonization of germ-free *A. thaliana* roots with Root179 wild-type (WT) and mutant strains in an agar-based system revealed that 3 out of the 7 tested mutant strains were impaired in root colonization compared to the parental strain at 14 days post inoculation (Supplementary Fig. 10a, Kruskal–Wallis, $\chi^2 = 60.713$, $P < 0.001$). Pairwise comparisons using Dunn's test validated that *ΔpstABCS* ($P < 0.05$), *ΔtypA* ($P < 0.001$) and *ΔexbD1* ($P < 0.05$) mutants showed defects in root colonization compared to control WT Root179 based on live colony counts (Fig. 6c), but retained their wild-type-like growth in liquid Tryptic Soy Broth medium (Fig. 6d, ANOVA followed by Tuckey HSD, $P > 0.05$). The results indicate that these genes are dispensable for in vitro growth but likely act as root colonization determinants. To further validate these results, we generated complementation lines and empty-vector controls for *ΔtypA and ΔexbD1* (*ΔtypA::typA*, *ΔtypA::eV*, *ΔexbD1::exbD1*, *ΔexbD1::eV*) but could not complement the *ΔpstABCS* mutant due to the large size of the deletion fragment (3.7 kb). Mutant complementation with the respective genes restored the ability of both strains to colonize plant roots based on colony counts (Kruskal–Wallis followed by Dunn's test comparing CFUs per gram of roots of WT *vs.* complemented strains, $P > 0.05$; Fig. 6f). Next, we performed re-colonization experiments in the FlowPot system with Root179 WT, *ΔpstABCS*, *ΔtypA* or *ΔexbD1* mutants in the absence or presence of the 105-member multi-kingdom SynCom (Fig. 6g, h) and quantified strain abundance in roots using qPCR-based specific detection of Root179 (see primer validation in Supplementary Fig. 10b). Notably, defect in root colonization was again observed for *ΔtypA* and *ΔpstABCS* (yet to a lesser extent for *ΔtypA* in mono-association experiments, $P = 0.1875$, Fig. 6g), indicating that this effect was robust, irrespective of the system, the quantification method or the presence of microbial competitors (Fig. 6h). However, this was not the case for the *ΔexbD1* for which bacterial growth defect at roots was

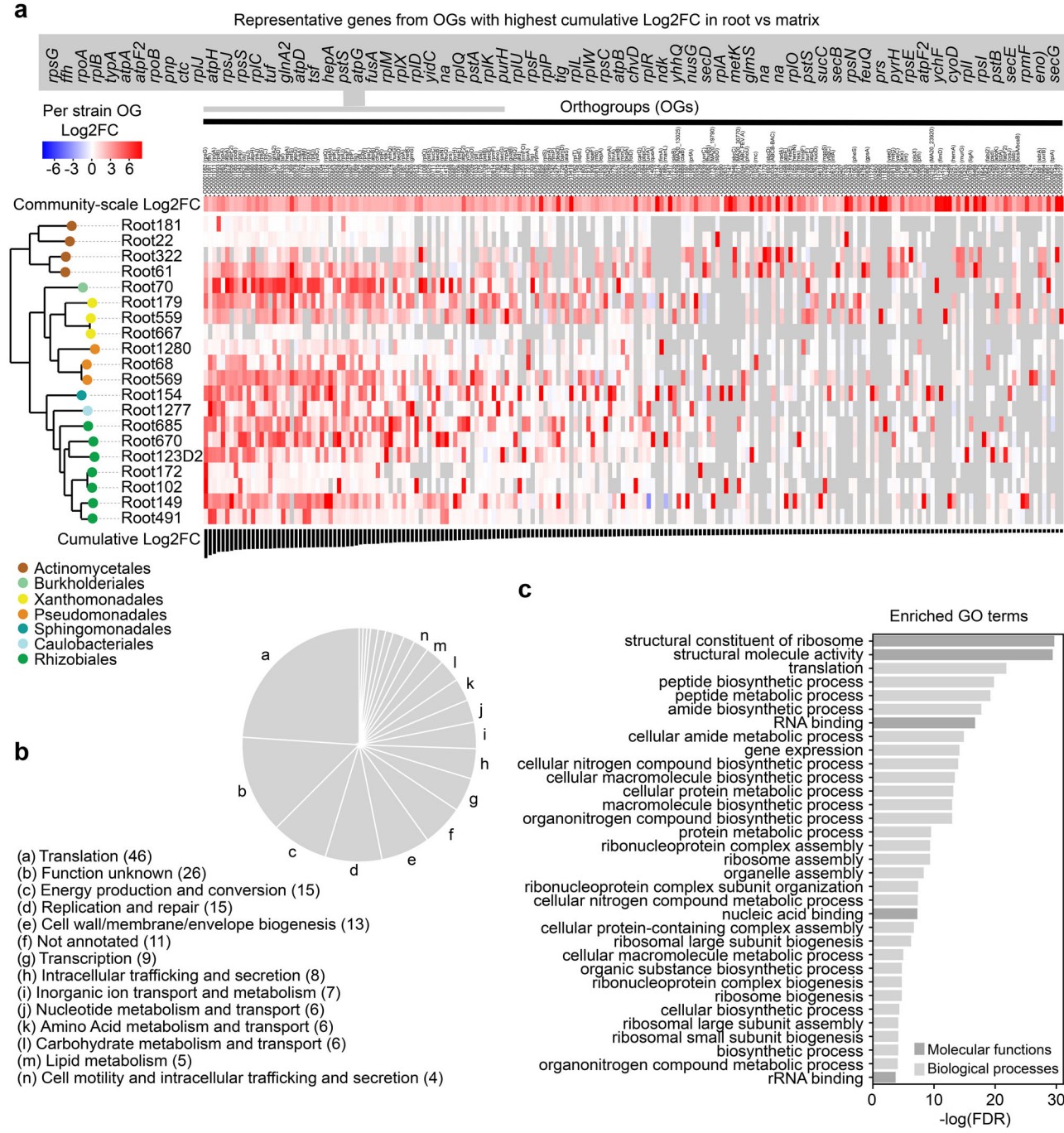

**Fig. 4 | Conserved gene families and functions activated by robust bacterial colonizers at roots. a** Heatmap showing orthogroups (OGs) differential expression in roots compared to soil matrix across the 20 most covered strains on roots. OGs presented are the top 200 showing the highest cumulative log2fc across all 20 bacterial strains and are ordered based on this cumulative log2fc (lower row). The names and annotation (when annotated) of the OGs are indicated (top). The log2fc values at the community scale (calculated by pooling transcripts counts of each OG across all strains) are indicated on the upper row. A phylogenomic tree computed with STAG (as implemented in OrthoFinder[50]) on the 20 bacterial genomes is displayed on the left. **b** COG category assignment (according to EggNog annotation[73]) of the 200 OGs and the number of OGs in each category is indicated. **c** GO-term enrichment (computed with topGO[41]) within the 200 OGs, showing the adjusted *P* values of significantly enriched GO terms (FDR < 0.05) for the GO categories Biological process (light gray) and Molecular Function (dark gray). *P* values where obtained from Fisher's exact test with *P* value adjustment for False Discovery Rate. Source data are provided as a Source Data file.

system-dependent (Fig. 6g, h). Inspection of sequence conservation of *typA*, *exbD1*, *pstA*, *pstB*, *pstC*, and *pstS* across the bacterial tree of life (i.e., 3837 bacterial genomes from Levy and colleagues[23]) revealed broad, multi-phyla sequence conservation for *typA*, and *pst* operon-containing genes, whereas *exbD1* sequence conservation was primarily constrained to the phylum Proteobacteria (Supplementary Fig. 10b). Our data indicate that the robust root colonizer Root179 deploys multiple strategies to proliferate at roots, involving evolutionary-conserved genes such as *typA* and *pstABCS* that are reproducibly activated by multiple strains during root colonization.

## Discussion

Structural and functional architectures of root-associated microbial communities are driving forces modulating plant growth and

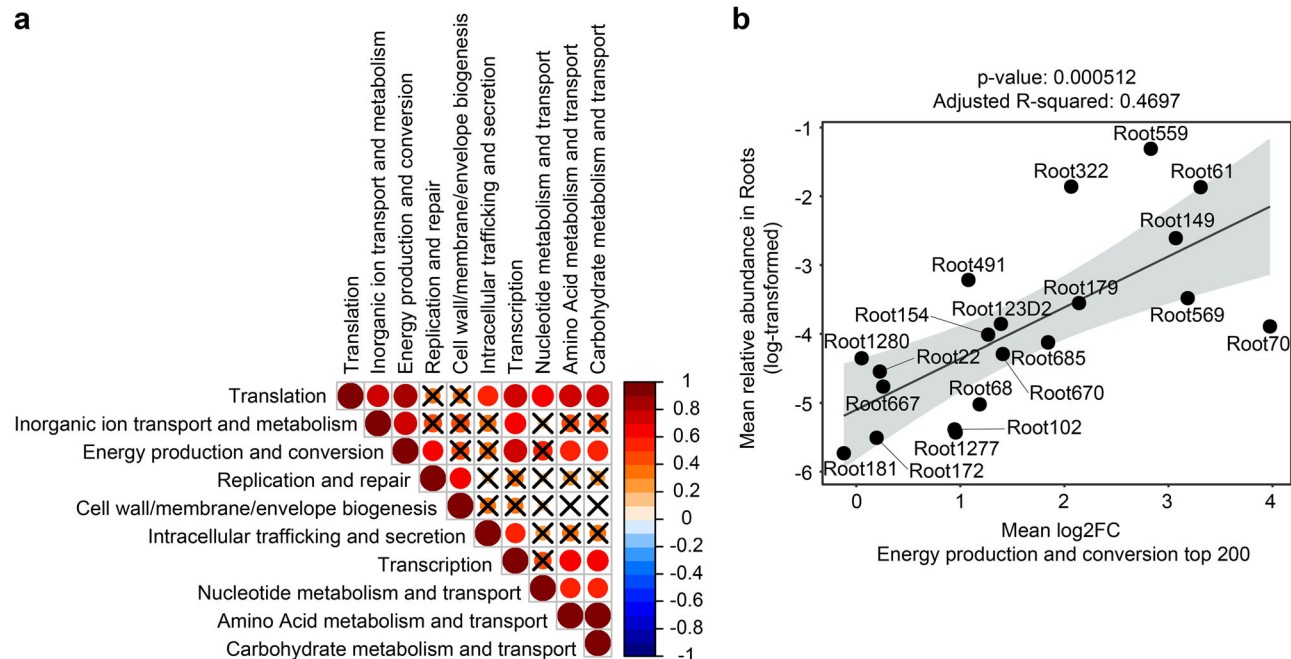

**Fig. 5 | Correlation between the expression of specific functions and relative abundance of strains in roots. a** Correlogram showing Pearson's correlations between functions regulation in the top-200 OGs in the roots (see Fig. 3a, b). Functions regulation is calculated as the mean log2FC of all OGs annotated with a given COG category in the top-200 OGs for each strain. Colors represent the Pearson's correlation value and nonsignificant correlations are crossed. **b** Linear regression between the mean log2FC of all OGs annotated as "Energy production and conversion" within the top-200 OGs and the log-transformed relative abundance of strains measured based on RNA read mapping (see Fig. 1a). Gray area represents the 0.95 confidence interval of the linear regression. Source data are provided as a Source Data file.

performance in nature. While our knowledge of the genomic features driving microbial adaptation to plant roots has rapidly progressed[22–26], our understanding of microbial processes and functions activated in complex microbial communities during root colonization remains fragmentary. Using a combination of multi-kingdom SynCom and reference genome-guided metatranscriptomic profiling, we successfully identified and experimentally validated several evolutionary-conserved genetic determinants that promote bacterial colonization at roots. Our results indicate extensive, yet partly conserved, functional adjustments to the root environment in phylogenetically distant microbial strains.

Compared to other methods used to identify root colonization determinants in microbes[22–26], reference genome-guided metatranscriptomics of SynComs allows identifying genes activated in a multi-kingdom community context at both strain- and community-level resolution during root microbiota establishment. The method however exclusively relies on low-diversity microbial communities assembled with culturable microbes. Despite the (i) development of strictly controlled microbiota reconstitution experiments with a low-complexity SynCom, (ii) successful depletion of rRNA from samples, (iii) deep Illumina-based RNA sequencing, and (iv) reference genome-guided mapping of Illumina short reads, we could not capture the entire SynCom metatranscriptome. Alternative strategies will be particularly needed to deplete plant-derived mRNA reads such as fractionation methods that can be used to separate microbial cells from plant cells[33], hybridization methods that could be used to efficiently deplete unwanted host-derived cDNA[52], or real-time selective RNA sequencing that emerges as a technique of choice to selectively sequence reads of interest from complex host-associated microbiota samples[53,54]. Our work nonetheless demonstrates that unraveling complex gene regulatory circuits activated within complex multi-kingdom microbial consortia in planta becomes realistic through this reference genome-guided metatranscriptomics approach.

Consistent with previously identified root colonization determinants[27], our results highlight that individual bacterial strains activate genetic programs involved in chemotaxis towards the roots (e.g., *cheY* or *tlpA*), movement, and primary attachment to the root surface (e.g., *fliC, pilA, oprF*) and to the root hairs (e.g., *pssA*) as well as secondary attachment and microcolony/biofilm formation (e.g., *pssA, lapA, rapA, uppS*). Our results also overlap with previous results from comparative genomics and RB-TnSeq/IN-seq studies[23,24,26], with many genes involved in transcription/translation, signal transduction or sugar and amino acids metabolism (especially Leucine metabolism) particularly activated upon root contact.

In this work, we experimentally validated the importance of two genes (*typA, exbD*) and a full operon (*pstABCS*) of Root179 that act as root colonization determinants. Both *typA* and *pst* genes (*A, B, C,* and *S*) are among the most commonly upregulated genes in the bacterial community, while *exbD* was specifically upregulated in Root179. The gene *typA* encodes a GTP-binding protein with a conserved sequence in most bacterial phylogenetic groups, has been previously identified[26] and independently validated[25] as required for root colonization in *Rhizobium leguminosarum* and *Herbaspirillum seropedicae* SmR1, respectively. This gene has also been identified in other host-associated bacteria, for example in the coordination of pathogenicity island expression in enteropathogenic *Escherichia coli*, in antimicrobial resistance and virulence in *Pseudomonas aeruginosa*[55], *Escherichia coli*[56] and in *Bordetella pertussis* infectious cycle[57]. ExbD is a biopolymer transport protein with a conserved sequence in most bacterial phylogenetic groups except in Actinobacteria and Bacillales (Supplementary Fig. 10b). This gene confers a selective advantage for root colonization in *Herbaspirillum seropedicae* SmR1[25]. ExbD is part of the energy-transducing device comprising the proteins TonB and ExbB required for active transport of mainly iron, heme and $Fe^{3+}$−siderophore complexes across the outer bacterial membrane. The role of ExbD has been mainly studied in *Escherichia coli* K-12, but similar transport systems

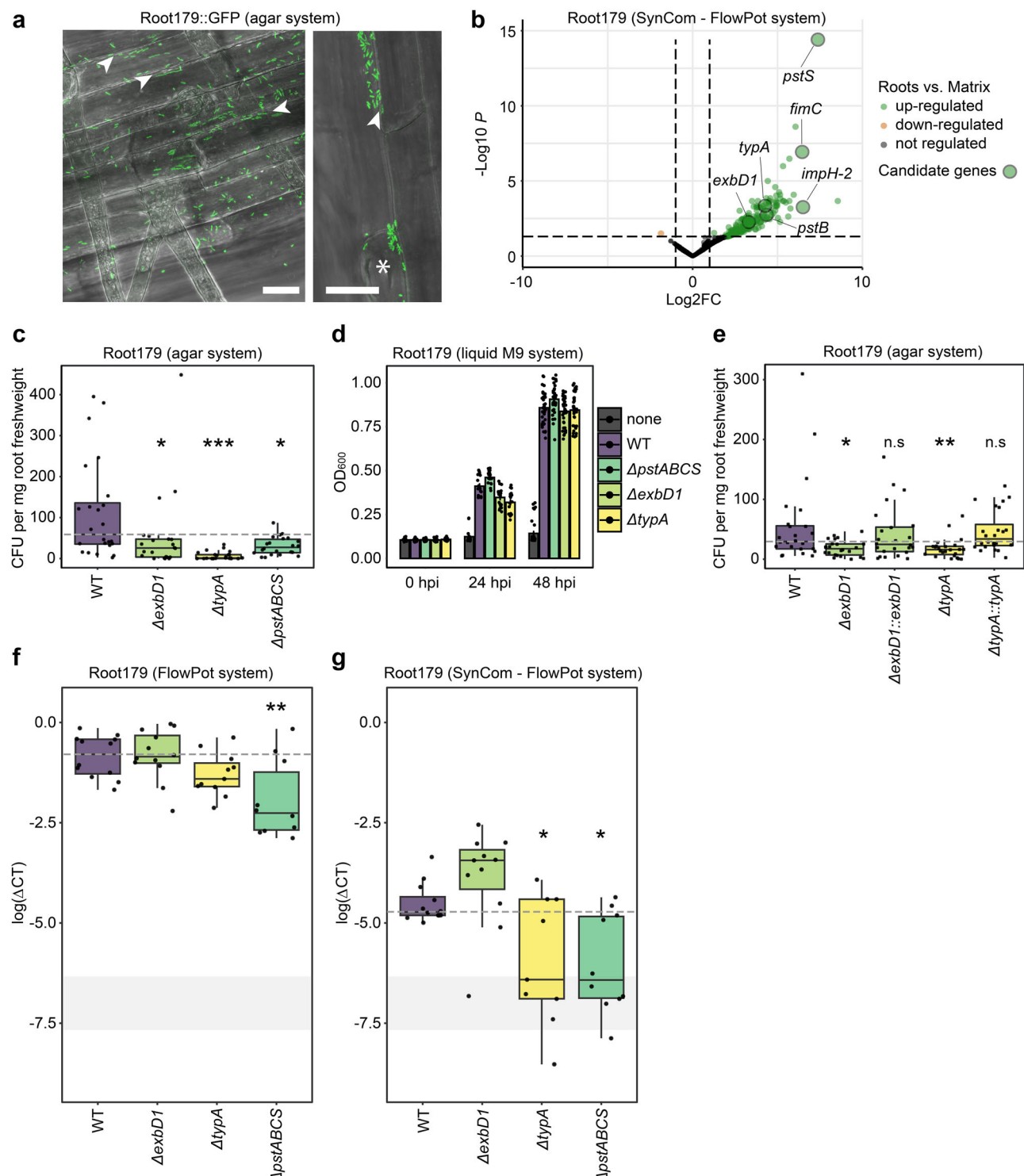

have been identified in different bacterial species, in particular those that infect humans and colonize plants[58–60]. ExbD has been recently studied in *Bacteroides thetaiotaomicron* and results indicate that it is involved in the uptake and utilization of polysaccharides[61]. Finally, we report that the entire *pstABCS* operon that encodes a phosphate transport system (but not *pstS* or *pstB* alone) promotes Root179 root colonization, suggesting that efficient phosphate uptake is required for successful root colonization. The different genes coding for the *pstABCS* operon are mostly conserved in bacterial phylogenetic groups except for Acinetobacter for which variations in *pstA*, *pstC*, and *pstS* can be observed compared to Root179 genes. This operon (and more

largely the Pho regulon) is involved in the response to the phosphate level at the cell membrane surface and connects phosphate sensing with bacterial response to environmental conditions and virulence[62–64].

Taken together, our results support the idea that bacterial activity is especially important at roots and that bacterial adaptation to nutrient availability (*pst* regulon for phosphate sensing and *exbD* for iron import) and overall abiotic conditions (response to stresses mediated by *typA* and the *pst* regulon) in roots is key for bacterial establishment. The fact that the *pst* operon (part of the *Pho* regulon) and *typA* have been described as bacterial virulence and pathogenicity determinants[56,57,62–64] in different hosts further corroborates their

**Fig. 6 | Validation of candidate genes involved in root colonization of *A. thaliana* in *Rhodanobacter* Root179. a** Fluorescence microscopy of GFP-labeled *Rhodanobacter* sp. Root179 (Root179::GFP) colonizing *A. thaliana* root surface at 14 dpi. Scale: 20 μm, Arrows: accumulation of bacteria in grooves between epidermal cells, Asterisk: root hair emergence site. This experiment was done only one time with no replicate. **b** Volcano plot depicting differentially expressed genes (DEGs) in Root179 between root and matrix samples (DESeq2, Wald test with FDR adjusted $P < 0.05$). DEGs are highlighted in color with transcripts significantly downregulated in roots vs. matrix in orange and upregulated in roots vs. matrix in green ($-1.5 < \log2FC > 1.5$). **c** Root colonization ability of Root179 WT and corresponding deletion mutants. Colonization is expressed by the number of colony-forming units (CFUs) per milligram of fresh *A. thaliana* roots grown in an agar plate-based system. Each strain was inoculated within MS medium supplemented with MES buffer and *A. thaliana* seeds were added to the plates. Plants were harvested at 14dpi. Roots were crushed and dilution series of lysed roots were plated on 50%TSB square plates for colony counting. *P* values were obtained from a Dunn's test following a Kruskal–Wallis test. *P* values between mutants and the WT strain were adjusted with FDR. $n = 24$ root samples for each treatment over two independent experiments. $P_{\Delta ExbD \text{ - } Root179WT} = 0.00975$, $P_{\Delta pstABCS \text{ - } Root179WT} = 0.02290$, $P_{\Delta ExbD \text{ - } Root179WT} < 0.0001$: *$P < 0.05$, ***$P < 0.001$. **d** OD$_{600}$ of WT and mutant strains impaired in root colonization ability in liquid cultures. $n = 18$ samples for each treatment over three independent experiments. **e** Root colonization ability of WT, knockout mutants and complemented Root179 strains. Colonization is expressed by the number of CFUs per milligram

of fresh roots (see above). *P* values were obtained from a Dunn's test following a Kruskal–Wallis test. *P* values between mutants and WT strain were adjusted with FDR. $n = 24$ root samples in total for each treatment over three independent experiments. $P_{\Delta ExbD::ExbD \text{ - } Root179WT} = 0.2586667$, $P_{\Delta ExbD \text{ - } Root179WT} = 0.0058$, $P_{\Delta typA::typA \text{ - } Root179WT} = 0.3093$, $P_{\Delta typA \text{ - } Root179WT} = 0.00440001$. ns nonsignificant, **$P < 0.0$. **f, g** Root colonization ability of Root179 WT and corresponding deletion mutants in the FlowPot system in the absence (**f**) and presence (**g**) of the multi-kingdom SynCom (shown in Fig. 1) using quantitative PCR, $n = 12$ total samples per treatment except for *ΔpstABCS* without syncom (**f**, $n = 10$) and for *ΔtypA* without syncom (**f**, $n = 11$). qPCR-based abundance of Root179 WT and mutants in roots is represented in log($\Delta CT$). Differences between each mutant and the WT were tested with a two-sided ANOVA followed by two-sided TukeyHSD test and the *P* values were adjusted with FDR. *$P < 0.05$, **$P < 0.01$; *P* in absence (**f**) of the SynCom were $P_{\Delta typA \text{ - } Root179WT} = 0.1875$, $P_{\Delta pstABCS \text{ - } Root179WT} = 0.0018$, $P_{\Delta ExbD \text{ - } Root179WT} = 0.9041$. *P* in the presence (**g**) of the SynCom were $P_{\Delta typA \text{ - } Root179WT} = 0.0014$, $P_{\Delta pstABCS \text{ - } Root179WT} = 0.0054$, $P_{\Delta ExbD \text{ - } Root179WT} = 0.2397$. The gray rectangle depicts the background qPCR noise detected for mock-inoculated root samples. The lower and upper hinges correspond to the first and third quartiles. The upper whisker extends from the hinge to the largest value no further than 1.5 * IQR from the hinge (where IQR is the interquartile range, or distance between the first and third quartiles). The lower whisker extends from the hinge to the smallest value at most 1.5 * IQR of the hinge. Source data are provided as a Source Data file.

potential involvement in plant-microbe interactions and communication. Finally, we note that the bacterial genes identified herein are important for the colonization of non-plant hosts in other bacteria (see Lamarche and colleagues[62] for a review on *pst* operon and virulence), suggesting that they might represent general features of microbial adjustment to the abiotic conditions within their hosts. This study provides community-level insights into the complex regulatory circuits and functions deployed by phylogenetically diverse root microbiota members to colonize roots of healthy plants.

## Methods

### Experimental design and sampling

FlowPots preparation. The FlowPot experiment was set up following the previously described protocol in Duran and colleagues[13] and Kremer and colleagues[38]. Briefly, the 84 bacterial strains (see list in Supplementary Data 6) were cultivated from glycerol stocks first on TSB + agar (Tryptic Soy Broth, Sigma) for a week and the 22 fungal strains (see list in Supplementary Data 6) were cultivated from glycerol stocks on PGA (Potato Glucose Agar, Sigma-Aldrich) for 2 weeks. Microbial mixtures were adjusted to a biomass ratio of 4:1 (eukaryotes:prokaryotes, as assessed by Joergensen and Emmerling, 2006[65]), using 200 μL bacterial inoculum and 200 μL of fungal inocula in 50 mL 1/2 MS (Murashige + Skoog Medium including Vitamins, Duchefa) supplemented with MES buffer (2-(*N*-morpholino)ethanesulfonic acid), which were then inoculated into the FlowPot using a 50 mL syringe. An identical procedure has been used for the second reconstitution experiment, except that the 84-member bacterial SynCom was inoculated alone or in the presence of the 22 fungi. Before inoculation, *A. thaliana* Col-0 seeds were sterilized using ethanol and bleach and stratified for 4 days in the dark at 4 °C. Ten seeds were sown per pot and the closed boxes incubated at 21 °C, for 10 h with light (intensity 4) at 19 °C and 14 h in the dark for 4 weeks. After 4 weeks of FlowPot incubation, roots were harvested from planted pots and peat matrix was sampled from unplanted pots. The roots of all plants from three FlowPots were combined for a total of nine plants per sample and six total samples, three from matrix and 3 from root. Roots were thoroughly washed in water, dried and frozen in liquid nitrogen and stored at −80 °C. Roots and peat matrix samples were homogenized in 2 mL Eppendorf screw cap tubes using the Precellys 24 tissue lyzer (Bertin Technologies, Montigny-le-Bretonneux, France) after deep freezing twice at 6200 rpm for 30 s. DNA and RNA were co-extracted

from both peat matrix and roots using the RNeasy PowerSoil Total RNA Kit (Qiagen, Hilden, Germany) and its extension for co-extraction of DNA. The RNA precipitation step temperature was adjusted to −20 °C for root samples.

### DNA samples processing

The concentration of DNA samples was fluorescently quantified, diluted to 3.5 ng/μL, and used in a two-step PCR amplification protocol. In the first step, V4–V7 of bacterial 16 S rRNA (799 F–1192 R) and fungal ITS1 (ITS1F - ITS2) were amplified (primers previously published in Getzke and colleagues[15]). Each sample was amplified in a 25 μl reaction volume containing 2 U DFS-Taq DNA polymerase, 1x incomplete buffer (both Bioron GmbH, Ludwigshafen, Germany), 2 mM MgCl$_2$, 0.3% BSA, 0.2 mM dNTPs (Life Technologies GmbH, Darmstadt, Germany) and 0.3 μM forward and reverse primers. PCR was performed using the same parameters for all primer pairs (94 °C/ 2 min, 94 °C/30 s, 55 °C/30 s, 72 °C/30 s, 72 °C/10 min for 25 cycles). Afterward, single-stranded DNA and proteins were digested by adding 1 μl of Antarctic phosphatase, 1 μl Exonuclease I and 2.44 μl Antarctic Phosphatase buffer (New England BioLabs GmbH, Frankfurt, Germany) to 20 μl of the pooled PCR product. Samples were incubated at 37 °C for 30 min and enzymes were deactivated at 85 °C for 15 min. Samples were centrifuged for 10 min at 2000× *g* and 3 μl of this reaction were used for a second PCR, performed as described above with only 10 cycles and with primers including barcodes and Illumina adapters. Afterward, the replicated reactions were combined and purified: (1) bacterial amplicons were purified using the QIAquick gel extraction kit (QIAGEN, Hilden, Germany); (2) fungal amplicons were purified using Agencourt AMPure XP beads. Each library was then purified and re-concentrated twice with Agencourt AMPure XP beads, and 100 ng of each library were pooled together. Paired-end Illumina sequencing was performed in-house using the MiSeq sequencer and custom sequencing primers.

### 16 S rRNA gene and ITS reads processing

Paired 16 S rRNA amplicon sequencing reads were joined (join_paired_ends QIIME, default) and then quality-filtered and demultiplexed (split_libraries_fastq, QIIME, with max. barcode errors 1 and phred score of 30)[66]. The filtered reads were dereplicated (usearch –derep_fulllength)[67], sorted by copy number (only reads >2 copies were retained) and were checked for chimeras using usearch (usearch

−uchime)[68]. All retained reads were aligned to the mapped against the reference 16S rRNA and ITS from genomes using PyNAST[69]; those that did not align were removed.

## RNA samples processing

The quality and concentration of RNA samples were quantified using a Bioanalyzer (Agilent, Waldbronn, Germany). Plant, bacterial and fungal ribosomal RNA were depleted from the roots samples using a combination of Illumina Ribo-Zero rRNA Removal Kit (Bacteria), Ribo-Zero rRNA Removal Kit (Yeast) and Ribo-Zero rRNA Removal Kit (Plant Seed/Root)(Illumina, San Diego, USA) following the manufacturer's protocol while only bacterial and fungal ribosomal RNA were depleted the for matrix samples. The quality and quantity were then quantified again using a Bioanalyzer (Agilent, Waldbronn, Germany). Library prep was done using the SMARTer Stranded RNA-Seq Kit (Takara Bio IncSaint-Germain-en-Laye, France) following the manufacturer's protocol. The quality and quantity were then quantified again using a Bioanalyzer (Agilent, Waldbronn, Germany). Illumina sequencing was then performed in-house using the HiSeq3000 sequencer and custom sequencing primers.

## Reference genomes and bioinformatics

To perform a reference-based transcriptomic analysis, we used the CDS sequences associated with each predicted gene in 84 bacterial and 22 fungal genomes, as well as in the *A. thaliana* TAIR10 genome (downloaded from http://arabidopsis.org). All bacterial genomes were previously published (downloaded from http://at-sphere.com)[37]. 17/22 fungal genomes were previously published (downloaded from https://mycocosm.jgi.doe.gov)[38]. The other fungi (5/22, Plectospaerella_cucumerina_MPI-CAGE-AT-0143: Plecto143, Fusarium_oxysporum_MPI-CAGE-AT-0013: Fusarium13, Fusarium_redolens_MPI-CAGE-CH-0216: Fusarium216, Fusarium_oxysporum_MPI-CAGE-CH-0226: Fusarium226, Fusarium_equiseti_MPI-CAGE-CH-0233: Fusarium233, four from genus *Fusarium* and one from species *Plectosphaerella cucumerina*) used in our experiments were sequenced de novo and their assemblies can be accessed via ENA (PRJEB50298 and PRJEB61839). First, their genomic DNA was extracted from freshly growing mycelium using a modified CTAB-based protocol, as previously described[70]. Genomes were then assembled using CANU v1.8[71], and assemblies were polished with Arrow v0.14. Gene prediction was performed with FGE-NESH v8.0.0[72], with reference matrix *Fusarium* for the 4 *Fusarium* strains and matrix *Torrubiella hemipterigena* for the *Plectosphaerella cucumerina* strain.

Fungal gene annotation was performed using emapper v2[73] but also different specialized tools: BUSCO2 v5.4.7[74] (with database ascomycota_odb10) to identify ascomycetal conserved genes likely involved in primary metabolism, SignalP3[42] v6.0 to predict genes encoding secreted proteins, EffectorP4 v3.0[43] (run in fungal mode on predicted secreted proteins) to identify candidate effector-encoding genes and dbCan2[44] v4.0.0 to annotate genes encoding carbohydrate-active enzymes. In addition, we used the tool PHIB-Blast (http://phiblast.phi-base.org/) to identify homologs of fungal differentially expressed genes that have been previously characterized and are referenced in database PHIBase6 v4.15[45].

Functional annotation of the microbial genomes was carried out using the tools emapper v2.1.5 and the EggNog database on the protein models associated with each predicted gene[73]. From the total EggNog annotation, we used COG categories, GO terms and KEGG orthology groups. To be able to compare the transcription of orthologous genes in the bacterial and fungal communities, two orthology predictions were performed with OrthoFinder v2.2.7[50], using default parameters. Functional annotation of OGs was performed by identifying the best representative sequence in each OG using HMMER v3.1 (as previously suggested by Mesny and colleagues[37]), and considering the annotation of this sequence as one of the total OG. Alternatively, to the use of

OrthoFinder-defined OGs, we also used the curated KEGG orthology groups, as implemented in the EggNog database and annotated by emapper. The bioinformatic pipeline used to analyze RNA datasets is summarized in Supplementary Fig. 11.

## Microbial transcriptome analysis

Quality check and trimming of resulting single-end RNA-Seq reads was performed using Trimmomatic v0.38[74] (options LEADING:20 TRAILING:20 AVGQUAL:20 HEADCROP:10 MINLEN:130). Reads were also deduplicated using USEARCH[67] (options -derep_fulllength -sizeout).

The pipeline of analysis used herein is represented in Supplementary Fig. 11. To associate the sequenced reads to genes in our reference genomes, we relied on the pseudo-mapping algorithm Salmon v0.14.1[75], that uses k-mer content to identify the most likely transcript of origin of each read. This high-throughput method was well suited to our dataset, considering the large number of reads sequenced and the size of our reference genome catalog. Prior to read pseudo-mapping, we used all gene coding sequences from the microbial and host genomes to build an index (with command *salmon index --perfectHash –keepDuplicates*). Since multiple genomes carry identical genes, we decided not to remove 100% similar sequences in the index (only later in the pipeline when analyzing individual strains transcriptomes), so reads are not falsely and arbitrarily associated to only one paralogous or orthologous gene. We then proceeded to the pseudo-mapping of each trimmed and deduplicated RNA-Seq sample individually by running *salmon quant --validateMappings --seqBias –incompatPrior 0.0* on our pre-computed index. Transcripts quantifications from the pseudo-mappings were then recovered and used in different analyses (described below and in Supplementary Fig. 11): (1) transcriptome-based estimation of strain abundances in roots and matrix; (2) per-strain gene differential expression analysis; (3) per-strain OG differential expression analysis; (4) OG differential expression analysis at the community scale. Besides the latter analysis, it was important to avoid that genes with identical sequences in multiple organisms introduce a bias in strain/transcript quantification. We therefore used a self-made script to remove interstrain duplicates from Salmon outputs, while keeping identical paralogs.

1. To calculate transcriptome-based strain relative abundance values, we calculated for each sample the percentage of reads associated to each strain. We considered a strain as detected when at least one read mapped the reference genome.

2. Per-strain differential expression analyses were performed independently for each strain using DeSeq2[39] and the exact same strategy was used for the second reconstitution experiment in order to compare DEGs. Because the use of DESeq2 requires (1) transcripts of a specific gene to be detected in all samples to be able to correct for sample coverage (i.e., normalize the counts) and (2) to detect enough transcripts per gene to compute a Wald test, we did not compute DESeq2 results for strains for which these two conditions were not met. We focused our main interpretations on the 20 most-abundant bacterial strains, to ensure that we only consider transcriptomes that were well covered by the sequencing. For all transcriptomes, per-strain transcripts per million (TPM) values were calculated and their distribution appeared to be normal or quasi-normal (Supplementary Fig. 4), confirming the accurate pseudo-mapping and the possibility to identify differentially expressed genes between roots and matrix. We also inspected the expression of six reference housekeeping genes that (i) were annotated by EggNog in our bacterial genomes; (ii) were present in a single copy in the genomes (no paralog); (iii) which expression has been characterized to be invariable in more than 80% of studies inspected by Rocha and colleagues[76]. We considered as differentially expressed all the genes which adjusted *P* values from DESeq2 are inferior to 0.05. In addition, we used the shrinkage algorithm

apeglm[77] to correct log2-transformed FC values, so differential expression is not over-estimated for transcripts with low read counts.

3. To perform comparative transcriptomic analyses and identify genes that are commonly differentially expressed in roots by bacteria, we conducted a per-strain OG differential expression analysis. We referred to our orthology prediction computed with OrthoFinder. The per-gene read counts obtained from Salmon pseudo-mapping were summed to obtain one read count per OG for each bacterial strain. The resulting aggregated read count values were used to perform independent differential expression analyses with DESeq2. As in (2), we considered as differentially expressed by one strain all the OGs with adjusted $P$ values from DESeq2 inferior to 0.05. In addition, we used shrinkage algorithm apeglm to correct log2-transformed FC values, so differential expression is not over-estimated for OGs with low read counts.

4. Finally, we analyzed OG differential gene expression at the community scale. We used unfiltered Salmon pseudo-mapping results and summed across all bacterial strains the read counts associated to genes clustered in one OG. We then performed a single differential expression analysis with DESeq2, using apeglm for log2-transformed FC value correction. Were considered as differentially expressed all the OGs which adjusted $P$ values were lower than 0.05.

As a control for (3) and (4), we also performed the same analyses using KEGG orthologous groups (as annotated by emapper v2)[73] instead of OGs predicted with OrthoFinder and observed overall similar results.

## Plant transcriptome analysis

To confirm the suitability of our data processing approach (i.e., pseudo-mapping of reads on a large multi-species genomic dataset), we also analyzed read counts associated to plant genes. We compared our read count values to those of a previously published RNA sequencing dataset of *A. thaliana* roots grown in the FlowPot system in the presence of a different synthetic community[9]. In this previous study, authors performed Poly-A enrichment prior to RNA sequencing and used a classical mapping algorithm with the *A. thaliana* genome as a single reference. Read counts resulting from this mapping were downloaded from GEO (GSE160106; file complete_table_3470.txt). Correlation of log-transformed mean read counts (over three samples) between this dataset and ours was performed using a Pearson correlation (computed using function stats.pearsonr from Python library Scipy). For this calculation, we only considered genes detected (mean read count >0) in both studies.

## DNA–RNA profiling comparison

To compare the SynCom profiles obtained from RNA- and DNA- seq we calculated relative abundances of strains based on 16 S/ITS read counts. We counted the number of strains detected by the two methods by considering a strain as detected whenever at least one read was detected in one sample. We then calculated the average relative abundance of each strain detected with the two methods and tested whether the abundances were correlated with a Spearman correlation test.

## Permutational analysis of variance of strains transcriptomes

To investigate whether the compartment and the taxonomy determined the individual transcriptomes within the metatranscriptome we used per-strain OG abundances tables (read counts for OGs of each strain in each sample). We used OGs read counts to be able to compare transcriptomes between strains. We calculated relative read counts of OGs for each strain in each sample and calculated distances between these individual OG-based transcriptomes using Bray–Curtis

distances. We represented PCOAs with these Bray–Curtis distances and used PERMANOVA analyzes of variance to test if different factors explained variance (i.e., distances) between individual transcriptomes. We computed different models for each taxonomic level to test whether some levels could better explain the variations between transcriptomes and computed models for roots and soil samples together or separately.

## Knockout mutant generation

**Gibson assembly.** Primers to amplify each two 750 bp-long flanking regions for every gene of interest were designed using Geneious Prime ® 2020.2 and in silico cloned into pK18mobsacB[78]. The plasmid was linearized via amplification. Per reaction, the master mix consisted of 10.8 μl nuclease-free water, 4 μl 5× Phusion HF buffer, 0.4 μl dNTPs (10 mM), each 1 μl forward and reverse primers (10 μM), 2 μl template, 0.2 μl Phusion DNA polymerase and 0.6 μl dimethyl sulfoxide (DMSO). The thermal cycles consisted in an initial denaturation at 98 °C for 30 s, followed by 35 cycles of denaturation at 98 °C for 7 s, primer annealing at 55 °C for 20 s, elongation at 72 °C for 15 s per kbp 150 s, and ending with a final elongation at 72 °C for 7 min. In order to remove the remaining circular pK18mobsacB plasmid, the PCR reaction was subsequently DpnI-treated. 1 μg of pK18mobsacB DNA was digested with 1 μl of DpnI restriction enzyme with 5 μl 10x NEBuffer, filled up to a total reaction volume of 50 μl with nuclease-free water. The vector was digested at 37 °C for 15 min and the enzyme was deactivated at 80 °C for 20 min. Afterward, the digestion of the vector was verified by loading 5 μl on a 1% agarose gel. The leftover linear pK18mobsacB was purified with Beckman Coulter's Ampure XP kit and the concentration was measured using a Nanodrop. Flanking regions from Root179 were PCR amplified from Root179 genomic DNA and subsequently purified. Plasmid backbone and flanking region inserts were fused using the Gibson assembly master mix (NEB) for 1 h.

**Transformation.** Chemically competent DH10 *E. coli* aliquots containing 50 μl of cells were mixed with 4 μl of the Gibson reaction. The cells were incubated on ice for 30 min, heat-shocked at 42 °C for 60 s, and subsequently chilled on ice for 2 min. Subsequently, 1 ml of 50% TSB medium was added, and the cells were allowed to regenerate at 37 °C for up to 1 h. The cells were then plated on TSA plates containing 25 μg/ml of Kanamycin and incubated overnight at 37 °C in order to select against non-transformed cells. The resulting colonies were verified by PCR and subsequent Sanger sequencing using plasmid-specific M13 primers (see Supplementary Information).

**Conjugation between transformed *E. coli* cells and Root179.** The plasmid harboring the two flanking regions was transformed into Root179 by tripartite conjugation with the plasmid containing DH10 *E. coli* and *E. coli* pRK600 as helper strain. The strains were mixed based on OD$_{600}$ measurements: Root179: *E. coli* transformant: *E. coli* pRK600 (2:1:1) and 50 μl of the conjugation mixture was spread on 50%TSA plates, followed by incubation at 25 °C overnight. The next day, the mating patches were washed of the plates with 1 ml of 50%TSB and diluted 1:100. The dilutions were then plated on 50%TSA containing 25 μg/ml of Kanamycin to select for the vector and 50 μg/ml Nitrofurantoin to select against the *E. coli* strains. The resulting colonies were verified by PCR and subsequent Sanger sequencing using a genome and a plasmid-specific primer.

**Sucrose counterselection.** Following the confirmation of plasmid integration, a single colony was inoculated into 1 ml of 50%TSB and vortexed. The number of cells was determined using the Multisizer 4e cell counter (Beckman) and four dilutions were made consisting of 50,000 cells/μl, 5000 cells/μl, 500 cells/μl. All dilutions were spread on plates containing 300 mM sucrose and incubated at 25 °C for 72 h.

The resulting colonies were verified by PCR and subsequent Sanger sequencing using a pair of genome-specific primers flanking the insertion site.

## Complementation of knockout mutants

Stable complementation lines of *ΔtypA* and *ΔexbD* were generated by genomically integrating the respective coding sequences expressed under a Kanamycin promoter into the genome of Root179 or mutants thereof using a Tn7 transposon-based integration. Cloning was conducted via Gibson assembly as described earlier; primers were designed using Geneious Prime ® 2020.2. The coding regions of the genes of interest were amplified from Root179 genomic DNA, the Kanamycin promoter was amplified from pK18mobsacB. A vector for site-specific integration mediated by Tn7 was constructed using Golden Gate-compatible Level 1 modules. Tn7 attachment sites, a gentamicin resistance cassette, and sacB selection marker were mobilized into Golden Gate-compatible L1 plasmids and subsequently assembled within a pSEVA211-based Golden Gate-compatible recipient by BpiI restriction and ligation. The plasmid backbone was linearized by amplification and fused to the inserts via Gibson assembly. As controls, empty vectors were designed that are solely composed of plasmid backbone and Kanamycin promoter, however, lack the target gene. Triparental mating was used to integrate the respective plasmids into Root179 with the help of a Tn7 transposase pTNS3 expressing *E. coli*. At each step, candidate colonies were verified by PCR and subsequent Sanger sequencing using plasmid or genome-specific primers.

## Confocal microscopy

The location of Root179 on *A. thaliana* roots was monitored by confocal laser scanning microscopy. To that end, an overnight liquid culture of Root179 expressing GFP under the control of the synthetic tac promoter[79] was harvested by centrifugation and washed twice in 10 mM MgSO4. GFP-tagged bacteria were inoculated into 0.5 MS (2.22 g.l−1 Murashige+Skoog basal salts, Sigma; 0.1 g.l−1 MES anhydrous, BioChemica; pH 5.7) at a final concentration of OD600 = 0.0005 by mixing bacteria and the medium prior to solidification. Approximately 15 sterile A. thaliana Col-0 seeds were sown on top and the plates were incubated vertically for 2 weeks (10 h light, 21 °C; 14 h dark, 19 °C). The root samples were visualized using a Zeiss LSM 880 inverted confocal laser scanning microscope with a LD C-Apochromat 40x/1.1 W Korr M27 objective lens, a Ch1: 493-598 filter and a 2.0% 488 nm laser. Maximum intensity projections were compiled from Z-stacks.

## Root colonization assays in agar plates

Bacterial mutants, inoculations, and plate incubation were done as described above for the microscopy. Roots were harvested after 2 weeks and washed with 10 mM MgSO$_4$, dried on autoclaved Whatman glass microfiber filters (GE Healthcare Life Sciences) and transferred to 2-mL tubes containing one stainless steel bead (3.2 mm). Roots were crushed using a Precellys 24 tissue lyzer (Bertin Technologies, Montigny-le-Bretonneux, France). Then 500 µl of 10 mM MgSO$_4$ was added and crushing was repeated. 10 µl of dilution series of lysed roots (1/1000, 1/100, and 1/10) were plated on 50%TSB square plates and incubated at 25 °C for 3 days. Colonies were then counted for each dilution.

## Growth assays in liquid media

The growth rates of the wild-type Root179 and the knockout mutants were measured using a Tecan microplate reader. Overnight liquid cultures of the bacteria were harvested by centrifuging 2 ml at 2000× *g* for 10 min. The supernatant was removed, the cell pellets were resuspended in sterile 10 mM MgSO$_4$ and the OD$_{600}$ of each bacterium was adjusted to 0.02. A 96-well plate (Greiner 96 flat, transparent) was then inoculated by adding 75 µl of resuspended bacteria to 75 µl of medium consisting of 50%TSB and using five technical replicates per bacterium, resulting in a concentration of 0.01 OD$_{600}$. Growth curves were done by measuring OD$_{600}$ in an absorbance plate reader every 10 min for 3 days at 19 °C with 10 s 250 rpm shaking every 5 min.

## Root colonization assays in flowpots

Inoculation of Root179 WT and mutants was done following the protocols used for both the bacterial and fungal SynCom inoculation. When co-inoculated with the multi-kingdom SynCom, Root179, and its mutants were introduced at equivalent concentration. Subsequent growth and harvesting procedures were executed as shown for the initial SynCom experiment. In total, 12 replicates were done per condition, consisting of two independent biological batches of 6 technical replicates. For root colonization measurement, root samples were frozen and crushed, total DNA was extracted using the FastDNA™ SPIN Kit for Soil (MPbio). Quantification of Root179 WT and mutants was performed through a quantitative PCR (qPCR) approach. Primers were designed to target Root179 genes that do not have homologs in other SynCom strains. Primers were designed using BatchPrimer3[80] (155-156-F: 5′-AGCCTTCAAGTCCTGACGAA-3′, 155-156-R: 5′-TTTGCTCACATCG CATGTTC-3′) their specificity was validated in silico utilizing the EMBOSS software[76,81]. Further validations of primers specificity and efficiency were conducted through PCR and qPCR on DNAs from R179 WT, mutants, and the bacterial and fungal SynComs. Normalization of qPCR results was performed by amplifying the *A. thaliana* Actin gene (actin2_short_F: 5′-atggaagctgctggaatccac-3′, actin2_short_R: 5′-ttgctc atacggtcagcgata-3′). Each qPCR reaction mixture consisted of 5 µL of iQ™ SYBR® Green Supermix (Bio-Rad), 0.3 µL of 10 µM Forward primers, 0.3 µL of 10 µM reverse primer, 1 µL of DNA, and MilliQ water, resulting in a final volume of 10 µL. The Bio-Rad CFX Connect Real-Time system was employed with the following cycling program: an initial denaturation step at 95 °C for 3 min, followed by 39 cycles of denaturation at 95 °C for 15 s, annealing at 60 °C for 30 s, and extension at 72 °C for 30 s. The quantification of R179 abundance (ΔCT) for each sample was calculated as $2^{-\Delta Cp}$.

## Conservation of root competence genes across the bacterial tree of life

To study the conservation of these genes across the bacterial tree of life, we referred to a previously published dataset of 3837 genomes[23] classified into nine phylogenetic groups. Orthology prediction was previously computed for each of these phylogenetic groups independently, defining gene families in these clades. We looked for gene families for which hidden Markov models show similarity to our genes *typA*, *exbD1*, *pstA*, *pstB*, *pstC*, and *pstS*. To do so, we did a similarity search with hmmscan v3.3[82] using the protein sequences of Root179 as a query, and the gene family hidden Markov models as subjects. For graphical representation, the similarity score of the best hit in each clade was normalized to the top score obtained for one gene. To obtain a phylogenetic tree showing the relationships between the clades, we used phyloT (https://phylot.biobyte.de/) and the taxonomic IDs associated to each clade name.

## Statistical analyses

Differential expression of genes, OGs, and KEGGS were tested in R 4.0.3 with the package DESeq2 1.30.1[39] using shrinkage algorithm apeglm 3.14[77] to correct Log2FC values. Differential relative abundances at the order, class and strain levels were also tested using DESeq2. All volcano plots of gene differential expression and strains/class/order abundance were done in R with the package EnhancedVolcano 1.11.3 using DESeq2 outputs. Pearson correlations between functions' average differential expression and strains abundance in the root compartment were done in R 4.0.3 using the corrplot function from the corrplot package (v0.92). Linear regressions between

functions' average differential expression and strains abundance in the root compartment were done in R 4.0.3 using the lm function in the package stats (v4.0.5). Colonization abilities of wild-type, mutant and complemented strains measured by colony counts were tested using a Kruskal–Wallis test followed by a Dunn's test in R, P values were then adjusted using the P.adjust function in the package stats v3.6.2. Colonization ability of wild-type and mutant strains measured by qPCR in flowpots in single or SynCom inoculations were tested using an ANOVA after log-transformation of the data. A subsequent TukerHSD test with a comparison matrix (using the emmeans and contrast functions in R) were used to test for differences between each mutant and the wild type. To perform Gene Ontology (GO) enrichment analyses, the software topGO v2.34.0[41] performing Fisher's exact tests) was used using EggNog GO annotations as an input and the Benjamini–Hochberg method for P value correction (FDR). A GO term was considered enriched in a gene set of interest when its FDR was lower than 0.05.

### Reporting summary

Further information on research design is available in the Nature Portfolio Reporting Summary linked to this article.

## Data availability

The community profiling data generated in this study have been deposited in the ENA database under accession code "PRJEB61839" (Root samples 16SrRNA: "ERS15411484", "ERS15411485", "ERS15411486"; Soil samples 16SrRNA: "ERS15411487", "ERS15411488", "ERS15411489"; Root samples ITS: "ERS15411490", "ERS15411491", "ERS15411492"; Soil samples ITS: "ERS15411493", "ERS15411494", "ERS15411495"). The genome of Plectospaerella_cucumerina_MPI-CAGE-AT-0143 generated in this study have been deposited in the ENA database under project ID "PRJEB61839" under sample ID "ERS15396667". The genomes of Fusarium_oxysporum_MPI-CAGE-AT-0013 (Fusarium13), Fusarium_redolens_MPI-CAGE-CH-0216 (Fusarium216), Fusarium_oxysporum_MPI-CAGE-CH-0226 (Fusarium226), Fusarium_equiseti_MPI-CAGE-CH-0233 (Fusarium233) have been deposited in the ENA database under accession code "PRJEB50298". The other genomes have been previously published and were obtained from "TAIR [http://arabidopsis.org]" (A. thaliana), "At-Sphere [http://at-sphere.com]" (82 bacteria), "Mycocosm [https://mycocosm.jgi.doe.gov]" (17/22 fungi). The raw RNA-seq reads, as well as the reference CDS used for mapping are available at GEO, under accession "GSE231841". All the data generated in this study are provided in the Supplementary Information/Source Data file. Source data are provided with this paper.

## Code availability

The scripts for the metatranscriptomic analysis pipeline from raw read processing to figure assembly (except differential expression analysis) are available at https://github.com/fantin-mesny/Scripts-from-Vannier-et-al.-2023, https://doi.org/10.5281/zenodo.10069685. The scripts for differential expression analysis and DNA community profiling are available at https://github.com/nathanvannierinrae/Scripts-from-Vannier-et-al.-2023.

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

## Acknowledgements

This work was supported by funds to S.H. from a European Research Council starting grant (MICRORULES 758003), the 'Priority Programme: Deconstruction and Reconstruction of the Plant Microbiota (SPP DECRyPT 2125)' and the Cluster of Excellence on Plant Sciences (CEPLAS), both funded by the Deutsche Forschungsgemeinschaft. We thank Paul Schulze-Lefert for providing feedback regarding this work.

## Author contributions

S.H. coordinated and supervised the project. N.V. performed all the experimental work, with support from L.D. and F.G. for bacterial mutagenesis. G.C. performed mutant inoculation experiments with SynComs. J.O. contributed to biological materials. T.T. provided support with amplicon sequencing data. N.V. and F.M. jointly analyzed transcriptome datasets. N.V. and S.H. wrote the manuscript, with input from F.M. and F.G.

## Funding

## Competing interests

The authors declare no competing interests.
