## [Peer Review File · Nature Communications]

Genome-resolved metatranscriptomics reveals conserved root colonization determinants in a synthetic microbiotaReviewer #1 (Remarks to the Author):

In this manuscript, Vennier et al. have undertaken in planta metatranscriptomics of various bacteria and fungi in a synthetic microbial community at strain resolution. This is a significant task, given the technical challenges associated with in planta metatranscriptomics of the microbiota, particularly bacteria, due to the abundance of host and microbial rRNAs. The authors' use of a genome-sequenced SynCom has facilitated a deep analysis of the transcriptomes of numerous strains within a taxonomically diverse community, although not all members of the SynCom could be analyzed. They have concurrently examined the changes in community structure and individual strain transcriptomes between soil and root, identifying microbial genes that respond during plant colonization. Furthermore, they have demonstrated that genes activated upon host contact are crucial for bacterial proliferation at roots.

Overall, this study delves into a significant topic in microbiota research, and its methodology has the potential to inspire future studies in the field. However, it appears that the full potential of the dataset has not been fully harnessed, thereby diminishing the true significance of their approach.

While the biological findings reported in this study may not be groundbreaking, I recognize the importance and quality of the dataset. I view this study as a valuable proof-of-concept for the methodology employed. However, I believe the paper could be enhanced by fully exploring the potential of the dataset and the approach. In my opinion, the notable strengths of their approach include (1) the simultaneous detection of "multi-kingdom" (bacteria, fungi, and plants) transcriptomes in the community context, and (2) strain-level resolution. I have provided specific comments below.

==Major comments==

Major Comment 1: This study predominantly analyzes bacterial transcriptome data, despite the dataset also containing fungal and plant transcriptomes. The study could be significantly enhanced by emphasizing the insights that can be gleaned from this multi-kingdom transcriptome data. A more in-depth analysis of the fungal data would be beneficial. I acknowledge that the experimental design (soil vs. plant) may not facilitate a meaningful analysis of the plant transcriptome. However, the authors could compare the plant transcriptome data obtained with their approach to that obtained with conventional plant RNA-seq to assess the quality of their plant transcriptome data. It would have been more insightful if the authors had incorporated another condition, such as a bacteria-only community (without fungi). This would enable the authors to analyze how the presence of fungi impacts bacterial metatranscriptomes and host transcriptomes.

Major Comment 2: The ability to generate strain-aware metatranscriptome data is a significant strength of the dataset produced in this study. However, the study does not offer substantial insights into how the presence of other microbes influences the transcriptomes of individual strains. One potential solution could be to analyze the transcriptomes of one or more strains (possibly the more abundant ones) in a mono-association setup and compare these results with those obtained in the SynCom experiment. This additional experiment would also enable the authors to meaningfully analyze plant transcriptome data, as suggested in Major Comment 1.

Major Comment 3: Related to Major Comment 2, could the authors comment on comparisons between relatively closely related strains? Did closely related strains show similar responses in planta (e.g., at the OG or GO term level)?

Major Comment 4: Most analyses (such as those presented in Fig4) are based on FC. How does the "baseline" expression (in this case, expression in the soil) compare between these strains? Could a low mean FC be due to high gene expression in the soil? The authors might consider calculating Transcripts Per Million (TPM) separately for soil and plant.

Major Comment 5: The authors restricted their analyses to the 20 most abundant bacterial strains due to limited coverage. I'm curious if the authors could aggregate less abundant strains for a more in-depth analysis. How does the metatranscriptome of less abundant strains compare to that

of highly abundant strains? Is there lower expression of "Energy production and conversion" OG in less abundant strains?

Major Comment 6: The authors identified the 200 most commonly induced bacterial OGs based on cumulative FCs. I'm wondering if this could be influenced by a few strains that express the OG at an exceptionally high level. In Figure 3A, there are OGs that are induced in only a few strains (somewhere in the middle of the heatmap); those OGs seem to be absent in many strains (indicated by the gray color). Would it be reasonable to classify such OGs as "commonly induced OGs?"

==Minor comments==

Minor Comment 1: In FigS1b, I presume that colors other than black represent rRNA, but this is not explicitly stated in either the legend or the figure panel.

Minor Comment 2: How variable is rRNA depletion performance? Showing replicate information in FigS1b would be helpful.

Minor Comment 3: Fig3a: gene names are hard to read. Could the authors show representatives with larger font size?

Minor Comment 4: A discussion on the remaining challenges and limitations of the method presented in this study would provide valuable insights for the readers.

Reviewer #2 (Remarks to the Author):

Genome-resolved metatranscriptomics reveals conserved root colonization determinants in a synthetic microbiota.

This paper is quite straightforward in flow. The authors inoculate roots of gnotobiotic *A. thaliana* plants with a diverse mixture of bacterial and fungal cultivated microbes, and then after a period of growth they extract DNA from soil and from whole roots for analysis of community composition via 16S / ITS marker genes, and they extract RNA from the same samples for metatranscriptomics. They remove ribosomal rRNA from the samples, sequence, and then look for genes that are up or downregulated in roots vs. the soil in the whole community. Among the genes they identify, they knock them out from one of their bacterial strains and test the effect on the mutant. Some of them affect growth in roots, but not in liquid growth media.

In principle the work seems solid, but as the paper currently is written there are some major omissions regarding the bioinformatics pipeline that make it difficult to determine how those steps are operating, and how trustworthy those steps are. Potentially the authors can resolve these major issues with a careful rewriting of the text. I highlight my issues with the paper below, roughly in order of appearance in the text.

--- Section 1 results

> Can explicitly state somewhere that there are 6 total samples, 3 from matrix and 3 from root, each a pool of 9 plants.

--- "53 out of 84 bacterial genomes"

> Are the genomes to which no reads mapped also those which were much less abundant by 16S / ITS sequencing? Are there any notable inconsistencies where there were lots of 16S / ITS reads but no RNA, or vice versa?

--- Line 133 "We then aggregated read counts at the order level for both bacteria and fungi to

make RA profiles comparable"

> Say instead, "to make it possible to compare RA profiles at the same taxonomic level". In popular language, calling two things "comparable" is like calling them "similar", and as currently worded, it sounds like you are applying a transformation to force the RA profiles themselves to be similar, when instead you are just converting the taxonomic level so you can compare apples to apples. It could be a problem with my way of reading, but I wouldn't be confused if you made the suggested change.

--- Line 140

> What is the actual test used for these P values?

--- Line 141 "whereas a significant change in ___ and ___ was only detected by RNA-based"

> Why could this be? Can lower metabolic activity in this group be a reason? What is the reasoning to trust RNA more than DNA?

--- Line 150

> What kinds of thresholds are used for including strains in DeSeq? Does a bacterium or fungi need a certain proportion of its genome covered by reads in order to qualify for DeSeq analysis? Does the coverage hitting common housekeeping genes in each genome need to be consistent with a normal distribution, or anything like this? How do I know that there weren't some strains that attracted many reads of certain categories in an unbalanced fashion.. like high coverage for some housekeeping genes but not others that one also would expect to find... issues such as this? Can you do some analysis to show that you are mapping reads more or less as expected, and not just relying on the algorithm to handle it correctly?

--- Line 162

> Related to above, was there a minimum number of reads that had to be assigned to a genome in order to look for DEGs? For example, if a genome needs 100,000 reads to be included, then even though a particular gene of interest in that genome is represented by 10,000 reads, that genome would be disqualified because of not enough total reads? The numbers I mention are not recommendations... just examples.

--- Line 150-165...

> How are the RNA reads normalized? Reads per million? These steps are done with Salmon, but what exactly does Salmon do?

--- Line 220 "and averaged it by GO-terms".

> It's unclear what this means... "averaged it"... please rewrite to be more explicit what is happening here.

--- Line 223 to 224 "different functions explain the same variance"

> Can you provide an example of a case in which you suspect this is happening?

--- Line 253 to 254 "retained their wild type-like growth in liquid TSB medium"... line 255 "but likely act as root colonization determinants"

> It would be especially relevant here if these mutants were tested in the same environment as the rest of the experiments... that is, do they maintain their growth in soil matrix but not in roots grown in the soil matrix. It's a possibility the bacteria may have lost abilities to grow in nutrient-poor conditions, for example, but perhaps nothing plant-specific as their growth in soil matrix might also be affected. If the authors wish not to do such an experiment, the authors should further tame the language and say "but could/may act as root colonization determinants" (don't say "likely")

--- Line 283 "allows to identify genes"

> this phrasing occurred once elsewhere.. correct grammar should be "allows one to identify genes" or "allows identifying genes"

--- Line 447 "Transcriptomic analysis.

> When determining if an organism is "detected" in the analysis... what is the threshold? If it is

found in 1 of 3 reps is it detected? 2 of 3 reps? What read count in what number of reps? This overlaps with a previous concern I raised

>What is a duplicate gene? What are inter-strain identical genes? How (quantitatively ... sequence identify for example) are duplicates / identical genes distinguished from orthogroups?

FIGURES

--- Fig 1.

> Panel a is too small. Would be great to see the names of the strains in the phylogenetic tree, but can barely see in the print version. Also, would be nice to be able to trace things horizontally across the panels. Can you add tick marks every 5 or 10 rows for example, to each section, so it's possible to follow each strain horizontally across the panels by eye?

> Panel c shows that sample 2, matrix, RNA has the most Actinomycetales, but sample 2, matrix, DNA, has among the least. How to explain this? Conversely, Actinomycetales have a much higher relative abundance considering RNA than DNA. How to explain? Also, in such analysis, has variation in 16S / ITS copy number in strains been considered? Why or why not? In fungi, there can be very many ITS copies. Without copy number correction, 16S / ITS quantification could overrepresent the abundances of some strains.

> Panel d, sample 2, roots in RNA and sample 2, roots in DNA have a vastly different population of Mortierellales. Can the authors speculate as to why? Is it a technical artifact of some method?

---Fig 2.

> Would be nice that there is a color legend in panel (a) that says orange is downregulated and green is up. I know it's in the legend, but would be so easy and appreciated to put that info in the fig.

> would be nice if in panel (c) the legend took the same format as (b). That is, matrix and roots are shown with black circles and triangles in the upper right, as in (b), and then below the chart the same info is repeated, but with a colored box. Also in (b), to avoid confusion, the color legend below the chart could have boxes instead of circles, since circle are supposed to represent "matrix"

---Fig 3

> would be nice if the tree had colored dots at the tips (as was done in Fig 1a), as the colored branches are hard to see.

Reviewer #3 (Remarks to the Author):

The work of Vannier et al. aims to reveal the genetic determinants of plant-microbe interactions using the matrix-root interface as a model in *Arabidopsis thaliana*. To do this, they used a multi-kingdom synthetic community with bacterial and fungal members in the FlowPot system to apparently overcome the technical difficulties associated with metatranscriptomics. The manuscript identifies microbial transcriptional changes upon root colonization. This transcriptional reprogramming included the expression of genes related to the processes of translation and energy production which seems to be a common strategy used by microbes to readapt their metabolism to the metabolic restrictions imposed by root niches. Some of the identified genes were validated by mutagenesis and functional complementation in the bacterium *Rhodanobacter* sp. (Root179 in the manuscript). The manuscript is well written and some of the observations are interesting. However, I would like to share with the authors my comments on the manuscript.

Line 129. "RNA-based abundance profiling revealed changes in relative abundance (RA) profiles at strain level resolution." I think in this case should be a DNA-based abundance profile, the RNA-

based approach does not have this level of resolution.

Line 131. "with eight bacterial and four fungal strains significantly enriched in roots vs. matrix and seven bacteria and a single fungal strain showing the opposite pattern". Could the authors comment on this very low level of enrichment? I think it compromises the main conclusions of the manuscript that it is ultimately based on the analysis of gene expression in a very small number of microbes.

I suggest showing in the manuscript the actual number of bacteria and fungi detected by the DNA-based method. I cannot find this information in the manuscript and it is relevant to assess the robustness of the conclusions.

The authors showed that, in the case of fungi, the proposed method failed to uncover the genetic determinants of plant colonization. How good was the multi-kingdom synthetic community design in this case? Is this system ecologically relevant?

Line 176 "To conclude, although the root and soil compartments are adjacent, extensive transcriptional reprogramming was observed across multiple independent strains, indicating major functional shifts during root microbiota establishment". I think this conclusion is overstated, based on the data presented, only a very small number of bacteria (14/84, and no changes was reported in fungi) changed their transcriptional responses in the plant root. Among these 14 bacteria, only 4 of them showed significant gene repression. I don't see this major functional shift during root microbiota establishment, in the data presented.

Line 209 "Together, our data indicate that a substantial fraction of bacterial processes induced at roots involve evolutionary-conserved functions that promote translation, energy production and transport, reflecting active bacterial growth and associated high protein biosynthesis rate in the root niche". This conclusion is not novel and it is predictable to some extent, others previous works have shown this conclusion before (doi: 10.15252/embr.202255380; doi: 10.1371/journal.pbio.2002860. doi:10.1038/s41588-017-0012-9).

Line228 "The results indicate that in planta activation of these processes is strongly linked with bacterial proliferation at roots and therefore suggests a possible link between the ability of diverse bacteria to utilize root-derived cues for their primary metabolism and their ability to dominate in the root microbiota". The link between activation of bacterial metabolic processes and bacterial proliferation is not demonstrated in this work. The ability of the root microbiota to utilize root-derived compounds have been amply demonstrated in previous works, as well as the bacterial ability to dominate in the community as a condition for root colonization.

The author validated 3 genes using the bacterium *Rhodanobacter* sp. (Root179) as determinants of root colonization in mono-association assays. I was wondering if these genes are also relevant in the community context? This will emphasize the main conclusions of this manuscript.

Do the authors believe that the peat matrix used could influence the final results of this work? Given that bacterial root colonization ability changes in response to environmental fluctuations, how robust are the validated gene functions under different abiotic conditions?

In general, I believe that this manuscript does not represent a step forward in the identification of new bacterial genetic determinants of plant colonization and is not novel enough to be published in this journal.

REVIEWER COMMENTS

Reviewer #1 (Remarks to the Author):

In this manuscript, Vennier et al. have undertaken in planta metatranscriptomics of various bacteria and fungi in a synthetic microbial community at strain resolution. This is a significant task, given the technical challenges associated with in planta metatranscriptomics of the microbiota, particularly bacteria, due to the abundance of host and microbial rRNAs. The authors' use of a genome-sequenced SynCom has facilitated a deep analysis of the transcriptomes of numerous strains within a taxonomically diverse community, although not all members of the SynCom could be analyzed. They have concurrently examined the changes in community structure and individual strain transcriptomes between soil and root, identifying microbial genes that respond during plant colonization. Furthermore, they have demonstrated that genes activated upon host contact are crucial for bacterial proliferation at roots.

We thank the reviewer for the positive feedback and for the great suggestions.

Overall, this study delves into a significant topic in microbiota research, and its methodology has the potential to inspire future studies in the field. However, it appears that the full potential of the dataset has not been fully harnessed, thereby diminishing the true significance of their approach.

While the biological findings reported in this study may not be groundbreaking, I recognize the importance and quality of the dataset. I view this study as a valuable proof-of-concept for the methodology employed. However, I believe the paper could be enhanced by fully exploring the potential of the dataset and the approach. In my opinion, the notable strengths of their approach include (1) the simultaneous detection of "multi-kingdom" (bacteria, fungi, and plants) transcriptomes in the community context, and (2) strain-level resolution. I have provided specific comments below.

We agree that the method has more potential than we initially thought. We have performed new analyses to scrutinize host and fungal expression data, and performed new experiments to compare bacterial transcriptional reprogramming in response to fungi and in response to the host. We also provide a more in-depth comparison of RNA read-based and classical amplicon-based community profiling. These new data have extensively improved the quality of our work and reveal now the full potential of the method (see below).

==Major comments==

Major Comment 1: This study predominantly analyzes bacterial transcriptome data, despite the dataset also containing fungal and plant transcriptomes. The study could be significantly enhanced by emphasizing the insights that can be gleaned from this multi-kingdom transcriptome data. A more in-depth analysis of the fungal data would be beneficial.

We fully agree. We have now more thoroughly analyzed fungal genes differentially expressed in planta (DEGs, n = 94). These DEGs are associated to a broad diversity of functions and did not show any significant functional enrichment (topGO P > 0.05). Annotation of these DEGs revealed a significant proportion (20/94) of Ascomycetal core genes likely involved in primary metabolism (BUSCO2), a similar number of genes predicted to encode secreted proteins (18/94; SignalP3), four encoding candidate effector proteins (SignalP3+EffectorP4) and eight encode carbohydrate active enzymes (dbCan5) (See updated Supplementary Table 3). Interestingly, 11/94 genes have well conserved homologs that were previously studied in the context of host-fungi interactions (PHIBase6). Overall, this new analysis highlighted the over-expression in planta of fungal genes involved in general metabolism, but also of some genes likely involved in host colonization (i.e., secreted effectors, cerato-platanin secreted proteins, peptidases like subtilisin and proteases, homologs of Mes17 and BbAcs18; Supplementary Table 3 and lines 169-176).

I acknowledge that the experimental design (soil vs. plant) may not facilitate a meaningful analysis of the plant transcriptome. However, the authors could compare the plant transcriptome data obtained with their approach to that obtained with conventional plant RNA-seq to assess the quality of their plant transcriptome data.

This is a great point and we fully agree that we had initially overlooked the plant transcriptome data in the first version of the manuscript. As the reviewer mentioned, our experiment was not designed to analyze DEGs on the host side because we focused on microbial transcriptional reprogramming ex planta vs. in planta. This is because we looked at the host side in a recent study (Hou et al. 2021). However, we fully agree that it is indeed essential to validate that our plant expression data do make sense and to convince the reader that our method, which is based on rRNA depletion (and not mRNA enrichment), can also be used to simultaneously capture the host transcriptome.

We therefore provide a more detailed analysis of the root transcriptome (see Supplementary Fig. 1d) by comparing *A. thaliana* gene expression in roots (mean read count per gene) between our dataset and another dataset from the above-mentioned published study (Hou et al. 2021) in which we used the same gnotobiotic system, the same growth conditions and a slightly different multi-kingdom SynCom that contains >90% of the strains that we used here (referred to as normal light, NC+BFO in Hou et al. 2021). In root samples, we detected most *A. thaliana* genes as expressed and observed that 98% of these genes overlapped and showed consistent expression levels (Pearson, $r = 0.75$; $P < 0.0001$) with those of Hou et al. 2021 (see new Supplementary Fig 1d). This demonstrates that our method is suitable for simultaneous inspection of the multi-kingdom transcriptomes of the host and its associated bacterial and fungal microbiota. We make it clear in the text as well. See lines 119 to 128.

It would have been more insightful if the authors had incorporated another condition, such as a bacteria-only community (without fungi). This would enable the authors to analyze how the presence of fungi impacts bacterial metatranscriptomes and host transcriptomes.

We have carefully considered this suggestion and decided to perform a second reconstitution experiment with the 84-member bacterial SynCom inoculated in the presence or absence of the 22-member fungal SynCom in matrix samples in the absence of the host (See new Supplementary Fig. 6). Our primary objective here was to validate our method by testing whether bacterial DEGs regulated in response to the host (roots vs matrix) and in response to fungi (new experiment) differ or not. We identified 191 bacterial DEGs that responded to the fungal SynCom, contrasting with the 3,068 bacterial DEGs that responded to the presence of the host. Almost no overlap was observed between these two sets of DEGs (0.15%) whereas extensive overlap did exist between the sets of expressed genes (56.44%). From this analysis (Supplementary Fig. 6) we learned three important conclusions:

1: More bacterial genes are regulated in response to the host than in response to fungi. We can therefore validate our conclusion that "extensive bacterial transcriptional reprogramming occurs during root colonization".

2: Very different gene sets are induced by bacteria in response to the host and in response to fungi (0.15% overlap between the sets of DIFFERENTIALLY expressed genes), which validates our method.

3: The vast majority of bacterial genes are detected in a reproducible manner, irrespective of the presence of the host or the fungi (56.44% overlap between the sets of expressed genes).

We however think that presenting a detailed analysis regarding this new experiment is not justified (OGs, gene lists, GO terms, functional enrichment...). This is because 1) we want to make the manuscript focused on root colonization determinants, 2) we would not have the space to present all these data in a meaningful way, and 3) we think this information would deserve to be published in another article in which we will focus on bacterial-fungal competition in the root microbiome. If the reviewer insists, we can of course describe everything but we believe that it will negatively impact the flow of the manuscript and make it less straightforward to understand. We believe that the new Supplementary Fig. 6 represents a good compromise. See also lines 191 to 201.

Note that based on ref#2 and ref#3 comments, we have also performed a new experiment in the FlowPot system using Root179 WT and mutant strains either inoculated alone or with the 105-member SynCom. These new results indicate that defects in root colonization observed in the agar-based matrix (mono-association) were retained in the FlowPot system (microbial community context) for Root179 Δ PstABCS and Δ typA mutants, indicating that this effect is very robust, irrespective of the system, the method or the presence of microbial competitors. However, this was not the case for the Δ ExbD mutant for which a defect in root colonization

was observed in the agar-based matrix system but not in the FlowPot system. This indicates that the observed phenotype is system-dependent for this gene (See Fig. 6, Supplementary Fig. 10 and lines 292-300). See also response to ref#2 and ref#3 below.

Major Comment 2: The ability to generate strain-aware metatranscriptome data is a significant strength of the dataset produced in this study. However, the study does not offer substantial insights into how the presence of other microbes influences the transcriptomes of individual strains. One potential solution could be to analyze the transcriptomes of one or more strains (possibly the more abundant ones) in a mono-association setup and compare these results with those obtained in the SynCom experiment. This additional experiment would also enable the authors to meaningfully analyze plant transcriptome data, as suggested in Major Comment 1.

We have also very carefully considered this suggestion. Although we do agree that it would be relevant to compare bacterial gene expression in mono association vs. in a community context, we think that this experiment goes beyond the scope of the manuscript and would require a huge effort (cost-wise and time-wise). We could have done this experiment for one strain at most but we believe that this would remain superficial and would not be sufficient to extract the broad picture. We also believe that our new reconstitution experiment using the bacterial SynCom in the presence vs. absence of the fungal SynCom is already partly addressing the request (i.e., how the presence of other microbes influences the transcriptomes of individual strains).

Major Comment 3: Related to Major Comment 2, could the authors comment on comparisons between relatively closely related strains? Did closely related strains show similar responses in planta (e.g., at the OG or GO term level)?

That's an interesting question and we broaden the question to the link between taxonomy in general and transcriptomic programming. To address it, we performed new analyses using PERMANOVA (permutational analysis of variance, see New Supplementary Table 4 and See also lines 209 to 216) to test whether the taxonomy could explain variations in OG expression at different taxonomic levels (phylum, class, order, family, species) (see data as shown in the PCOA plot in Fig. 3b,c).

We observed that a significant fraction of the variance in bacterial transcriptomes (distance between OGs expression in the different samples) can be explained by the taxonomy (i.e., the taxonomic groups). Of course, because the genomes' content in OGs are correlated with the taxonomy, the more precise the taxonomy is the more variance between transcriptomes is explained (phylum~6%, class~13%, order 23%, family~40%, species ~55%). Around 6.7% of the variation in bacterial transcriptomes is explained by the "compartment" (roots vs matrix), which represents the part of the transcriptomes that varies in the same way between roots and matrix samples across all strains. Interestingly, a significant part of the variance is also explained by the interaction between "compartment" and "taxonomy" (and this at different taxonomic levels), highlighting that a significant proportion of the transcriptomic reprogramming is taxon specific (up to ~15% for the species and 11% for families). We also performed this analysis for roots and soil samples separately (to analyze the differences in expression between strains in each compartment). It shows that transcriptome variation explained by the "taxonomy" is higher between roots samples than between soil samples (for example in roots "species" explains 90% of variance while it is 64% in soil, "family" also is 65% and 47% respectively). Overall, these results indicate that:

- i) a large part of the transcriptomic adjustment to the root environment is strain specific and a fraction is common to all strains**
- ii) the more precise is the taxonomy considered the more variation in transcriptomes can be explained (less unexplained transcriptome variation at the species or family levels than at the phylum, class or order) and thus that more closely related strains have more similar transcriptomes**

Major Comment 4: Most analyses (such as those presented in Fig4) are based on FC. How does the "baseline" expression (in this case, expression in the soil) compare between these strains? Could a low mean FC be due to high gene expression in the soil? The authors might consider calculating Transcripts Per Million (TPM) separately for soil and plant.

We thank the reviewer for this comment, although we might not fully understand the question/concern. We shrank genes fold change values with the LFC shrinkage apeglm tool classically used with DESeq in order to avoid a bias linked to the baseline expression of genes (i.e. lower logFC because gene has high baseline expression and conversely high logFC because very low baseline expression). Secondly and to better describe the baseline expression of strains, we represented the frequency of logTPMs for the 20 strains in soil and plant samples (Supplementary fig 4, and lines 179-181 and 537-544). This new figure shows that for most strains the distribution of genes TPMs are distributed normally in soil and roots and are in the same order of magnitude in both compartments.

Major Comment 5: The authors restricted their analyses to the 20 most abundant bacterial strains due to limited coverage. I'm curious if the authors could aggregate less abundant strains for a more in-depth analysis. How does the metatranscriptome of less abundant strains compare to that of highly abundant strains? Is there lower expression of "Energy production and conversion" OG in less abundant strains?

Thanks very much for raising this point. We already reported in the text the total number of DEGs detected in all strains (3,068) and compared it to our selected set of 20 strains (2,899). Therefore, only 169 DEGs were detected in the strains that were not included in the top 20. As requested, we have now tested potential GO term enrichment in this list but no terms were found as enriched/depleted. There are too few differentially expressed OGs in this set and the coverage for these genes is too low to be able to correlate energy production and conversion to strains' RAs. Note that by focusing on the top 20 bacteria, we already looked at 95% of the dataset. However, we do agree that these genes are of potential importance and therefore we decided to include them in the Supplementary table 2 that now includes all DEGs from ALL strains.

Major Comment 6: The authors identified the 200 most commonly induced bacterial OGs based on cumulative FCs. I'm wondering if this could be influenced by a few strains that express the OG at an exceptionally high level. In Figure 3A, there are OGs that are induced in only a few strains (somewhere in the middle of the heatmap); those OGs seem to be absent in many strains (indicated by the gray color). Would it be reasonable to classify such OGs as "commonly induced OGs?"

We agree that strains that have extremely high OG FC values will influence the cumulative FC value more than strains that have low OG FC values. After checking other options (i.e., considering the average FC or not considering the FC values but only counting the number of strains showing differential OG enrichment), we realize that these options all have weaknesses. We therefore decided to keep it as it is and insist on the top OGs with the highest cumulative Log2FC that are induced by multiple strains (see lines 223-224, 228, 236 in the text; table title line 1500; gene list highlighted in grey in Fig. 4). Also, note that we do no longer call them "commonly-induced OGs" in the text but "OGs having highest cumulative Log2FC across strains" (see lines 223-224, 228, 236 in the text).

==Minor comments==

Minor Comment 1: In FigS1b, I presume that colors other than black represent rRNA, but this is not explicitly stated in either the legend or the figure panel.

Correct, the figure has been updated.

Minor Comment 2: How variable is rRNA depletion performance? Showing replicate information in FigS1b would be helpful.

We performed this control experiment before the "real" RNAseq experiment to precisely quantify rRNA depletion performance between total RNA samples and corresponding rRNA-depleted samples. We sent only one sample for each condition for sequencing because we consider this as a technical validation. Therefore, we only provide the depletion performance for a single replicate. Given that for the "real" experiment, all samples were prepared in the same way with the exact same protocol, we do not expect between-sample variation in rRNA depletion performance.

Minor Comment 3: Fig3a: gene names are hard to read. Could the authors show representatives with larger font size?

Thanks. We fully agree. The figure has been updated and representative gene names for OGs having highest cumulative Log2FC across strains are now visible.

Minor Comment 4: A discussion on the remaining challenges and limitations of the method presented in this study would provide valuable insights for the readers.

Thanks very much. We already included several sentences in the previous version that the reviewer might have overlooked.

“Despite the i) development of strictly controlled microbiota reconstitution experiments with a low-complexity SynCom, ii) successful depletion of rRNA from samples, iii) deep Illumina-based RNA sequencing, and iv) reference genome-guided mapping of Illumina short reads, we could not capture the entire SynCom metatranscriptome. Alternative strategies will be particularly needed to deplete plant-derived mRNA reads such as fractionation methods that can be used to separate microbial cells from plant cells, hybridization methods that could be used to efficiently deplete unwanted host-derived cDNA, or real-time selective RNA sequencing that emerges as a technique of choice to selectively sequence reads of interest from complex host-associated microbiota samples.”

We also make it clear now that “the method however relies on low-diversity microbial communities that are assembled using microbes that can be cultured.” If the reviewer has other specific weaknesses in mind beyond the fact that we used synthetic communities (cultured microbes, low diversity) and not natural communities, we will be happy to include this in the next version. See lines 324-325.

Reviewer #2 (Remarks to the Author):

Genome-resolved metatranscriptomics reveals conserved root colonization determinants in a synthetic microbiota.

This paper is quite straightforward in flow. The authors inoculate roots of gnotobiotic *A. thaliana* plants with a diverse mixture of bacterial and fungal cultivated microbes, and then after a period of growth they extract DNA from soil and from whole roots for analysis of community composition via 16S / ITS marker genes, and they extract RNA from the same samples for metatranscriptomics. They remove ribosomal rRNA from the samples, sequence, and then look for genes that are up or downregulated in roots vs. the soil in the whole community. Among the genes they identify, they knock them out from one of their bacterial strains and test the effect on the mutant. Some of them affect growth in roots, but not in liquid growth media.

In principle the work seems solid, but as the paper currently is written there are some major omissions regarding the bioinformatics pipeline that make it difficult to determine how those steps are operating, and how trustworthy those steps are. Potentially the authors can resolve these major issues with a careful rewriting of the text. I highlight my issues with the paper below, roughly in order of appearance in the text.

We thank the reviewer for the overall positive feedback. We now provide a much more detailed material and method section regarding this part and have updated the GitHub repository in which all steps and associated codes are much detailed (see also point-to-point responses below). See M&M section “Transcriptomic analysis” lines 502-579.

--- Section 1 results

> Can explicitly state somewhere that there are 6 total samples, 3 from matrix and 3 from root, each a pool of 9 plants.

We clarified this aspect in the method and added the number of samples for each analysis in the figure legends ($n = 6$) throughout the manuscript. See line 410.

--- “53 out of 84 bacterial genomes”

> Are the genomes to which no reads mapped also those which were much less abundant by 16S / ITS sequencing? Are there any notable inconsistencies where there were lots of 16S / ITS reads but no RNA, or vice versa?

We thank the reviewer for this important suggestion. We have now extensively worked on the comparison between RNA-based read mapping and DNA-based 16rRNA/ITS amplicon profiling and we decided to show this information in a new main figure (referred to as Fig. 2 in this second version of the manuscript). We now provide abundance barplots for both methods at different taxonomic levels (See supplementary Fig. 2) and tested correlations between DNA and RNA-based relative abundances at strain-level resolution (See Fig. 2e,f). We now report that DNA-based 16rRNA/ITS amplicon profiling detected 40 of the 84 strains of the SynCom in the roots and 41 in the matrix while RNA-based read mapping against reference genomes detected more strains: 64 in the roots and 68 in the matrix. Similarly, amplicon profiling identified 19 out of 22 fungi in roots and 20 in matrix whereas all fungi were identified by RNA-based profiling in both roots and matrix. We observed that a large proportion of the strains that have low relative abundance based on RNA-based read mapping ($RA < 0.1\%$) were not detected by amplicon sequencing (57% in roots, 55% in matrix). However, comparison between the two profiling methods revealed high overlap in the number of detected strains, with 46% of the bacteria (39 out of 84 for matrix) and 87% of the fungi (20 out of 22 for matrix) consistently identified using the two methods. See also lines 135-143 and 155-157.

Beyond the detection sensibility mentioned above, it is important to note that differences between the two methods can be very diverse, including differences in degradation/persistence of RNA/DNA in the environment, primer bias and 16SrRNA/ITS copy number variation, true strains variability in activity, genome quality (RNA), differences in sequencing depth (billions of reads for RNA few thousands for DNA), differences in bacterial cell types regarding DNA/RNA extraction/purification methods.

Irrespective of these differences, we are now confident to conclude that our method is adequate to capture diversity and composition of multi-kingdom microbial assemblages.

Also note that in the previous version of the manuscript we reported 53 genomes with reads mapped because we considered only genomes with multiple reads mapped in both roots and soil. To be consistent with the new RNA/DNA comparisons based on species with at least a single transcript detected in either soil or roots samples, we now report in the text 73 out of 84 genomes with reads mapped (see line 113).

--- Line 133 “We then aggregated read counts at the order level for both bacteria and fungi to make RA profiles comparable”

> Say instead, “to make it possible to compare RA profiles at the same taxonomic level”. In popular language, calling two things “comparable” is like calling them “similar”, and as currently worded, it sounds like you are applying a transformation to force the RA profiles themselves to be similar, when instead you are just converting the taxonomic level so you can compare apples to apples. It could be a problem with my way of reading, but I wouldn’t be confused if you made the suggested change.

We thank the reviewer for this improvement, we rephrased this section for clarity.

--- Line 140

> What is the actual test used for these P values?

We thank the reviewer for noticing that this information was missing. We specified that these P values were obtained from DESeq2 differential abundance testing in the figures and text and explained in the method section that DESeq2 is using a Wald test to calculate p-values.

--- Line 141 “whereas a significant change in ___ and ___ was only detected by RNA-based”

> Why could this be? Can lower metabolic activity in this group be a reason? What is the reasoning to trust RNA more than DNA?

As mentioned above, our RNA-based read mapping approach appear to detect more strains so it is likely that this discrepancy is, at least to some extent, explained by the higher strain detection rate. However, we cannot exclude the possibility that many other factors play a role,

including metabolic activity of course. As the reviewer might know, it is very difficult to determine causality here.

--- Line 150

> What kinds of thresholds are used for including strains in DeSeq? Does a bacterium or fungi need a certain proportion of its genome covered by reads in order to qualify for DeSeq analysis?

To test for differential enrichment DESeq2 requires 1) transcripts of at least one specific gene to be detected in all samples to be able to correct for sample coverage (i.e., normalize the counts) and 2) to detect enough transcripts per gene to obtain a distribution allowing to compute a Wald test. When testing for differential enrichment DESeq provides warnings if it fails to normalize the counts or to compute differential enrichment because no gene was detected in all samples or not enough transcripts were detected and DESeq cannot fit the distribution. Of course, the better the coverage the better the differential expression testing and that is why we focused on the top20 strains with the highest transcriptome coverage. Prior to computing DESeq2 enrichment testing we validated that read counts were normally or quasi-normally distributed for each of the 20 selected bacterial strains (see the response below with TPM distribution). Note that we still tested all transcriptomes for which DESeq2 could compute a differential expression test (we mention in the text the total number of DEGs in all strains).

Does the coverage hitting common housekeeping genes in each genome need to be consistent with a normal distribution, or anything like this? How do I know that there weren't some strains that attracted many reads of certain categories in an unbalanced fashion.. like high coverage for some housekeeping genes but not others that one also would expect to find... issues such as this? Can you do some analysis to show that you are mapping reads more or less as expected, and not just relying on the algorithm to handle it correctly?

Thanks very much. To address this important point, we calculated per-strain $\log(\text{TPM})$ values of bacterial detected genes, in matrix and roots independently. According to the distribution of these values (see histograms), read count are normally or quasi-normally distributed (except for Root667) (See new Supplementary Fig. 4 and lines 179-182 and 538-544). Pseudo-mapping results are therefore adapted to further proceed to a differential expression analysis using DESeq (see above). As suggested, we inspected the expression of bacterial housekeeping genes. We selected 6 reference housekeeping genes that (1) were annotated by EggNog in our bacterial genomes; (2) were present in a single copy in the genomes (no paralog); (3) showed constitutive expression in more than 80% of studies inspected by Rocha et al. (2015; 10.1007/s10482-015-0524-1). The TPM value of these housekeeping genes is shown by vertical bars on the histograms. For each gene in each of the 20 strains that we selected for in depth analysis, we show barplots of the $\log_2\text{FoldChange}$ values (roots vs matrix) and their standard error. The ones considered to be differentially expressed by DESeq2 ($\text{padj} < 0.05$) are indicated with an asterisk. Overall, housekeeping genes do not show differential expression in our dataset (See new Supplementary Fig. 4). We note that *rpoB* and *rho* were found differentially expressed in more than two strains. These genes were previously described to have constant expression in 86.1% and 82.6% of the studies, respectively (over a set of ~1,000 studies; see Rocha et al., 2015). Considering both the normal distribution of $\log(\text{TPM})$ values and the expression of housekeeping genes, we do believe that our dataset represent true bacterial gene expression, and is not heavily biased.

--- Line 162

> Related to above, was there a minimum number of reads that had to be assigned to a genome in order to look for DEGs? For example, if a genome needs 100,000 reads to be included, then even though a particular gene of interest in that genome is represented by 10,000 reads, that genome would be disqualified because of not enough total reads? The numbers I mention are not recommendations... just examples.

All strains' transcriptomes that could be tested using DESeq2 where tested. As stated above, DESeq2 needs transcripts detected in all samples to be able to correct for sample coverage (i.e. normalize the counts) and it needs to detect enough transcripts to compute a test based on a normal distribution. We included all strains ($n = 27$) for which DESeq2 was able to

compute a differential expression test. We actually reported in the text the total number of DEGs detected in all these strains (i.e., 3,068) and compared it to our set of 20 selected strains (i.e., 2,899). To avoid confusion, we now include all these DEGs in the Supplementary Table 2.

--- Line 150-165...

> How are the RNA reads normalized? Reads per million? These steps are done with Salmon, but what exactly does Salmon do?

We thank the reviewer for noticing that this part of our analysis pipeline was not correctly explained. We accordingly re-wrote the method section describing how we pre-processed the reads, how the reads were pseudo-mapped and quantified with Salmon (which indeed calculates TPM) and how we treated the reads post-Salmon (see lines 507-521). We actually needed to use reads that were not normalized to be able to compute differential enrichment testing with DESeq for individual strains. To do so, we recovered the transcript quantification from Salmon, separated the reads of individual strains and then used DESeq2 normalization to handle differences in depth between samples. All this process is now described in detail in the method section and the supplementary figure 11 and we also provide all the references for each tool we used, that describe how the algorithms handle the mapping/quantification/normalization.

--- Line 220 "and averaged it by GO-terms".

> It's unclear what this means... "averaged it"... please rewrite to be more explicit what is happening here.

We clarified that we calculated the average log²FC per GO term.

--- Line 223 to 224 "different functions explain the same variance"

> Can you provide an example of a case in which you suspect this is happening?

The more the expression of the functions are correlated, the more likely it is that they explain the same variance of species abundance. In that regard, "Amino Acid metabolism and transport" and "Carbohydrate metabolism and transport" are very highly correlated (>90%) and thus explain the same variance of species abundance.

--- Line 253 to 254 "retained their wild type-like growth in liquid TSB medium"... line 255 "but likely act as root colonization determinants"

> It would be especially relevant here if these mutants were tested in the same environment as the rest of the experiments... that is, do they maintain their growth in soil matrix but not in roots grown in the soil matrix. It's a possibility the bacteria may have lost abilities to grow in nutrient-poor conditions, for example, but perhaps nothing plant-specific as their growth in soil matrix might also be affected. If the authors wish not to do such an experiment, the authors should further tame the language and say "but could/may act as root colonization determinants" (don't say "likely")

We fully agree and we have now performed a new experiment in the FlowPot system using Root179 WT and mutant strains either inoculated alone or with the 105-member SynCom. We have designed and validated primers that specifically amplify Root179 and not the other SynCom members (See validation in Supplementary Fig. 10b). These new results indicate that defect in root colonization observed in the agar-based matrix (mono-association) were retained in the FlowPot system (microbial community context) for Root179 Δ PstABCS and Δ typA mutants, indicating that this effect is very robust, irrespective of the system or the presence of microbial competitors. However, this was not the case for the Δ ExbD mutant for which a defect in root colonization was observed in the agar-based matrix system but not in the FlowPot system. This indicates that the observed phenotype is system-dependent for this gene (See Fig. 6, Supplementary Fig.10 and lines 292-300 and 698-721). We consider this consistency remarkable given the major difference in growth systems (agar- vs peat-based), the method (CFU counts vs. qPCR), the biotic conditions (single strain vs. community context) and the fact that this experiment was done by a completely independent person in the lab.

Note however, that although our qPCR-based method was able to detect WT and mutant strains in roots, it largely failed at detecting them in matrix samples. In short, Root179 was not detected in most of the soil samples (i.e., no CT values, except for 14 out of > 100 samples).

Either the strains, including the WT, really did not survive in the soil (we think this is very unlikely as Root179 was detected in the soil in the first experiment and in previous work, see Hou et al., 2021), or there was potential technical issues (i.e., soil DNA extraction or inhibition of qPCR reactions).

--- Line 283 “allows to identify genes”

> this phrasing occurred once elsewhere.. correct grammar should be “allows one to identify genes” or “allows identifying genes”

Done

--- Line 447 “Transcriptomic analysis.

> When determining if an organism is “detected” in the analysis... what is the threshold? If it is found in 1 of 3 reps is it detected? 2 of 3 reps? What read count in what number of reps? This overlaps with a previous concern I raised

If a transcript mapping the reference genome of a given strain is found in any of the 3 reps then the organism is considered as detected. We understand that this is not a stringent selection but given the fact that we have the reference genomes of individual strains, we can unambiguously distinguish them with high precision.

>What is a duplicate gene? What are inter-strain identical genes? How (quantitatively ... sequence identify for example) are duplicates / identical genes distinguished from orthogroups?

When mentioning “duplicate genes” we refer to genes with 100% similarity and when these genes are from different genomes we refer to them as “inter-strain identical genes”. In our pipeline we kept these genes for the mapping step to not bias the mapping but removed them later on in the analysis of individual strains transcriptomes to avoid false differential expression due to abundance variation of other strains. We clarified this point at different places in the manuscript, see Supplementary Fig. 11 and the method section (lines 525-526). Regarding the quantitative difference between orthogroups and identical genes, it highly depends on orthogroups prediction, which is also based on sequence similarity. The orthology prediction consists in a blast between all pairs of sequences followed by a clustering. In the end some orthogroups might thus contain more sequences than others as well as various levels of similarity. There is no absolute quantitative difference between orthogroups and some orthogroups may contain 100% similar sequences.

FIGURES

--- Fig 1.

> Panel a is too small. Would be great to see the names of the strains in the phylogenetic tree, but can barely see in the print version. Also, would be nice to be able to trace things horizontally across the panels. Can you add tick marks every 5 or 10 rows for example, to each section, so it's possible to follow each strain horizontally across the panels by eye?

Thanks for the suggestion. The figure has been modified accordingly.

> Panel c shows that sample 2, matrix, RNA has the most Actinomycetales, but sample 2, matrix, DNA, has among the least. How to explain this? Conversely, Actinomycetales have a much higher relative abundance considering RNA than DNA. How to explain? Also, in such analysis, has variation in 16S / ITS copy number in strains been considered? Why or why not? In fungi, there can be very many ITS copies. Without copy number correction, 16S / ITS quantification could overrepresent the abundances of some strains.

We now present analyses comparing DNA- vs RNA-based profiling and demonstrated that overall, high similarity was observed in the number of strains detected and in the RA profiles. At strain-level resolution, we observed that strains' RAs (i.e. for those detected by both approaches) were highly correlated between DNA- vs RNA-based profiling (See new Fig. 2e,f). Of course, difference do exist in profiles can be explained by the factors that we listed above

and that include 16s rRNA/ITS copy number variation.

To assess if the copy number of 16S rRNA sequences in bacterial genomes could impact our quantification, we used the software barrnap v0.9 (<https://github.com/tseemann/barrnap>) to annotate 16S sequences in each genome assembly. Out of 84 bacteria, 82 carry a single 16S copy in their genome. The bacterium Root935 has in its genome one full 16S copy, and an additional truncated 16S fragment (28% of the expected size). The bacterium Root151 carries two 16S fragments (68% and 30% of the expected 16S size) on two different contigs, that may correspond to one single 16S copy that has not been correctly assembled. Therefore, we believe that the low variation in 16S copy numbers in our bacterial community unlikely impacted our quantification of relative abundances, especially since the two strains carrying two putative 16S copies are among the less abundant strains in soil and matrix.

Since the ITS copy number varies much more in fungal genomes, we were willing to design primers to target a single copy gene conserved among our fungal community. However, we failed at designing such primers, due to the fact that the community has a broad phylogenetic diversity. To assess the impact of ITS copy number on fungal relative abundance estimations, we previously conducted a qPCR experiment on 24 fungi with very different ITS copy numbers (Mesny et al., 2021 – see Peer Review file). We took care of using the same amount of template for each qPCR reaction (i.e., 1 ng of genomic DNA). While the Cq values varied in between fungal strains, we could not identify any significant link between Cq values and ITS copy number in fungal genomes (ANOVA Cq~ITScopynumber, P = 0.9).

To conclude, we unfortunately do not know what exactly explains the subtle differences between both approaches but we are confident (based on the data shown) that our RNAseq read mapping approach is highly adapted to capture the diversity and taxonomic structure of multi-kingdom microbial communities that colonize plant roots.

> Panel d, sample 2, roots in RNA and sample 2, roots in DNA have a vastly different population of Mortierellales. Can the authors speculate as to why? Is it a technical artifact of some method?

Thanks for highlighting this variation. Those are complex microbiota reconstruction experiments with > 100 microbial strains. It is known that stochasticity does exist during community assembly and therefore we are not surprised to see subtle sample-to-sample variation in the community profiles.

---Fig 2.

> Would be nice that there is a color legend in panel (a) that says orange is downregulated and green is up. I know it's in the legend, but would be so easy and appreciated to put that info in the fig.

> would be nice if in panel (c) the legend took the same format as (b). That is, matrix and roots are shown with black circles and triangles in the upper right, as in (b), and then below the chart the same info is repeated, but with a colored box. Also in (b), to avoid confusion, the color legend below the chart could have boxes instead of circles, since circle are supposed to represent "matrix"

Thanks very much. The figure has been modified accordingly

---Fig 3

> would be nice if the tree had colored dots at the tips (as was done in Fig 1a), as the colored branches are hard to see.

Done.

Reviewer #3 (Remarks to the Author):

The work of Vannier et al. aims to reveal the genetic determinants of plant-microbe interactions using the matrix-root interface as a model in *Arabidopsis thaliana*. To do this, they used a multi-kingdom synthetic community with bacterial and fungal members in the FlowPot system to apparently overcome the technical difficulties associated with metatranscriptomics. The manuscript identifies

microbial transcriptional changes upon root colonization. This transcriptional reprogramming included the expression of genes related to the processes of translation and energy production which seems to be a common strategy used by microbes to readapt their metabolism to the metabolic restrictions imposed by root niches. Some of the identified genes were validated by mutagenesis and functional complementation in the bacterium *Rhodanobacter* sp. (Root179 in the manuscript). The manuscript is well written and some of the observations are interesting.

Thanks for the constructive feedback and for the important suggestions raised. We have performed several new analyses and experiments that we believe have substantially improved the quality of our work and have addressed most of the criticism.

However, I would like to share with the authors my comments on the manuscript.

Line 129. "RNA-based abundance profiling revealed changes in relative abundance (RA) profiles at strain level resolution." I think in this case should be a DNA-based abundance profile, the RNA-based approach does not have this level of resolution.

We have now extensively worked on the comparison RNA- vs DNA-based profiling and we decided to show this information in a new main figure (referred to as Fig. 2 in this second version of the manuscript). We now provide abundance barplots at different taxonomic levels to compare the two approaches (See new supplementary Fig. 2) and tested correlations DNA/RNA-based abundances at strain-level resolution (See Fig. 2e,f). We now report that DNA-based approach detected 40 of the 84 bacterial strains of the SynCom in the roots and 41 in the matrix while RNA-based approach detected more strains: 64 in the roots and 68 in the matrix. Similarly, DNA-based profiling identified 19 out of 22 fungi in roots and 20 in matrix whereas ALL fungi were identified by RNA-based profiling in both roots and matrix. Notably, comparison between the two profiling methods revealed high overlap in the number of detected strains, with 46% of the bacteria (39 out of 84 for matrix) and 87% of the fungi (20 out of 22 for matrix) consistently identified using both DNA- and RNA-based profiling. Therefore, we conclude that our method appears actually more precise and allowed to identify 80% of the inoculated bacterial strains and 100% of the inoculated fungal strains. See also lines 135-143 and 155-157.

We do not understand the comment saying that "RNA-based approach does not have this level of resolution". Genome guided RNA-Seq read mapping against all 106 microbial reference genomes has much more power than inspecting a single locus such as done by amplicon sequencing. Our approach has the strain-level resolution because each read is mapped against the cognate reference genome, which allowed us to also inspect both strain relative abundance (i.e., cumulative read counts) AND transcriptional reprogramming at strain level resolution. Potentially our method could also distinguish strains that have similar 16s rRNA or ITS sequences, which makes it for sure more precise than classical amplicon-based profiling methods.

Line 131. "with eight bacterial and four fungal strains significantly enriched in roots vs. matrix and seven bacteria and a single fungal strain showing the opposite pattern". Could the authors comment on this very low level of enrichment? I think it compromises the main conclusions of the manuscript that it is ultimately based on the analysis of gene expression in a very small number of microbes.

We respectfully disagree with the comment. Our data indicate that the RA of 20 strains is affected by the presence of the host. That's 27% of the strains if we consider the strains that are detected after 5 weeks in the system. We provide a new analysis showing strains RAs aggregated at different taxonomic levels in roots and matrix samples (Supplementary Fig. 2). This clearly showed that the RA profiles differ between Root and Matrix samples, especially when our RNA-based profiling method is used. Furthermore, inspection of a manuscript from another lab that successfully used SynCom reconstitution experiments revealed very similar percentage of strain enrichment between soil and roots (see for example Lebeis et al. Science; 38-member SynCom, root-enriched: 6, root-depleted: 8). Although it is more difficult to compare, 27% of differentially enriched strains is actually much higher than the percentage of OTUs/ASVs detected as differentially altered in root vs soil in environmental microbiome studies (see Edwards et al. 9% or Lundberg et al. 14% for instance). Furthermore, strains can

have similar RA in roots vs. matrix but this does not mean that their gene expression profiles remain unchanged. We actually observed extensive transcriptional reprogramming for many strains that were not identified as differentially enriched in roots vs soil (i.e. see Root322, Root61, Root149 for instance).

I suggest showing in the manuscript the actual number of bacteria and fungi detected by the DNA-based method. I cannot find this information in the manuscript and it is relevant to assess the robustness of the conclusions.

This is indeed a very important suggestion and we have now performed this new analysis. This revealed high overlap in the number of detected strains, with 46% of the bacteria (39 out of 84) and 87% of the fungi (20 out of 22) consistently identified across methods in matrix samples (Fig. 2a,b). Similarly, 46% of the bacteria (39 out of 84) and 85% of the fungi (19 out of 22) were consistently identified in root samples (Fig. 2a,b). However, as mentioned above, RNA-Seq read mapping detected more strains than amplicon-based method. For instance: 80% of the bacteria and 100% of the fungi inoculated in matrix were detected, whereas these percentages dropped to 48% and 90% for DNA-based amplicon profiling. See also lines 135-143, 155-157, and 581-587).

The authors showed that, in the case of fungi, the proposed method failed to uncover the genetic determinants of plant colonization. How good was the multi-kingdom synthetic community design in this case? Is this system ecologically relevant?

We understand the concern here and we were also expecting more genes differentially regulated by fungi during root colonization. Our previous work revealed that *A. thaliana* is hosting fungi but the diversity is reduced compared to other plant species. We also observed that these fungi are kept in check and actually rarely abundantly colonize the root endosphere. This is because they are controlled by the combined action of bacterial commensals on one hand and by a Brassicaceae-specific immune defence branch involving glucosinolates (tryptophan-derived specialized antifungal molecules). In the presence of bacteria and of an intact host innate immune system (which is the case in this study), root colonization by fungi is extensively reduced and this is actually critical for *A. thaliana* survival. Please see our recent work Duran et al. 2018 Cell, Wolinska et al. 2021 PNAS.

The Synthetic community was built based on a decade of previous work in which we have described the root microbiota of *A. thaliana*, established culture collections that largely resemble naturally-occurring root microbiome, and compared natural site vs. culture collections to reconstitute synthetic microbiomes that are physiologically relevant (see Bai et al. Nature, Duran et al. Cell). The multi-kingdom synthetic community used here is adapted from (Duran et al. 2018, i.e. we basically kept strains that can be distinguished based on their 16S rRNA or ITS sequences in order to have strain-level resolution for the amplicon sequencing approach as well). This remains one of the most complex multi-kingdom microbial community that has been reconstituted to date. To address the concerns of the reviewers, we looked at the prevalence of the ITS sequences of the 22 selected fungi in natural population of *A. thaliana* across Europe (data from Thiergart et al., 2020). We obtained 100% match against ASV sequence tags for 21 of them and observed that the corresponding ASVs were highly prevalent in root samples across successive years. On average these fungi were found in 66% of the root samples across 17 sites in Europe, which is twice as high compared to the average of all fungi detected in roots (i.e., 37%, based on $n = 326$ fungal ASVs detected in roots). Note that 70% of the selected fungi were consecutively detected across 3 years and 22% across two years. We are fully confident that our rational design has been done very carefully and that our SynCom is ecologically-relevant.

See the table below for individual strain prevalence:

MPI	Strain	zOTU	% identity	Prevalence (%)	Years detected
MPI-CAGE-AT-0001	Verdah1	Zotu130	100	68,283%	3
MPI-CAGE-AT-0009	Chame1	Zotu927	88	46,667%	1
MPI-CAGE-AT-0013	Fusarium13	Zotu2	100	100,000%	3
MPI-CAGE-AT-0016	Plecuc1	Zotu8	100	100,000%	3
MPI-CAGE-AT-0021	Daces1	Zotu1	98	94,444%	3
MPI-CAGE-AT-0023	Fusre1	Zotu5	99	75,404%	3
MPI-CAGE-AT-0026	llyeu1	Zotu57	100	47,172%	3
MPI-CAGE-AA-0104	Morel_U14_1	Zotu6	100	71,212%	3
MPI-CAGE-AA-0113	Fuseq1	Zotu2656	100	66,667%	1
MPI-SDFR-AT-0129	Chafu1	no ITS1 Seq match			
MPI-CAGE-AT-0134	Neora1	Zotu32	100	63,737%	3
MPI-CAGE-AT-0135	Zalva1	Zotu878	100	42,475%	3
MPI-CAGE-AT-0143	Plecto143	Zotu8	99	100,000%	3
MPI-CAGE-AT-0147	Dacma1	Zotu1	100	94,444%	3
MPI-CAGE-CH-0201	Fusven1	Zotu179	100	59,091%	3
MPI-CAGE-CH-0216	Fusarium216	Zotu5	100	75,404%	3
MPI-CAGE-CH-0226	Fusarium226	Zotu11652	99	79,167%	2
MPI-CAGE-CH-0230	Mictri1	Zotu175	91	37,879%	2
MPI-CAGE-CH-0233	Fusarium233	Zotu20	100	31,212%	2
MPI-CAGE-CH-0235	Stael1	Zotu32	85	63,737%	3
MPI-CAGE-CH-0241	Cylol1	Zotu310	100	15,000%	2
MPI-CAGE-CH-0243	Denna1	Zotu357	100	78,333%	2

Line 176 “To conclude, although the root and soil compartments are adjacent, extensive transcriptional reprogramming was observed across multiple independent strains, indicating major functional shifts during root microbiota establishment”. I think this conclusion is overstated, based on the data presented, only a very small number of bacteria (14/84, and no changes was reported in fungi) changed their transcriptional responses in the plant root. Among these 14 bacteria, only 4 of them showed significant gene repression. I don't see this major functional shift during root microbiota establishment, in the data presented.

Note that some bacteria likely died in the system because 73 were actually detected at the end of the experiment in roots + matrix samples. Of these 73 strains, we detected 72,984 gene transcripts and identified > 3,000 DEGs that could be identified by DESeq2 and that belong to 27 strains (see new Supplementary Table 2). We have now performed new PERMANOVA analyses that show that 6.8% of the transcriptional differentiation between strains is explained by the compartment (See new Supplementary Table 4). We also previously presented a PcoA plot showing clear functional differentiation between root and matrix samples.

To address the reviewer criticism, we also performed a second reconstitution experiment with the 84-member bacterial SynCom inoculated in the presence or absence of the 22-member fungal SynCom in matrix samples in the absence of the host (See new Supplementary Fig. 6). Our primary objective here was to validate our method by testing whether the bacterial gene sets differentially regulated in response to the host (roots vs matrix, previous version of the manuscript) and in response to fungi (new experiment) differ or not. We identified only 191 bacterial DEGs that responded to the fungal SynCom, contrasting with the 3,068 bacterial DEGs that responded to the presence of the host. No overlap was observed between these two sets of DEGs (0.15%) whereas extensive overlap did exist between the sets of expressed genes (56.44%). From this analysis (Supplementary Fig. 6, See also lines 191-201) we learned three important conclusions:

1: More bacterial genes are regulated in response to the host than in response to fungi. We can therefore validate our conclusion that extensive bacterial transcriptional reprogramming occurs during root colonization”.

2: Very different gene sets are induced by bacteria in response to the host and in response to fungi (0.15% overlap between the sets of DIFFERENTIALLY expressed genes), which validates our method.

3: The vast majority of bacterial genes are detected in a reproducible manner, independent of the presence of the host or the presence of fungi (56.44% overlap between the sets of expressed genes).

We found it actually remarkable that although the soil and root compartment are adjacent, such a high number of DEGs could be identified.

Line 209 “Together, our data indicate that a substantial fraction of bacterial processes induced at

roots involve evolutionary-conserved functions that promote translation, energy production and transport, reflecting active bacterial growth and associated high protein biosynthesis rate in the root niche". This conclusion is not novel and it is predictable to some extent, others previous works have shown this conclusion before (doi: 10.15252/embr.202255380; doi: 10.1371/journal.pbio.2002860. doi:10.1038/s41588-017-0012-9).

While we agree with the reviewer that the regulation of functions involved in translation or energy production has been reported in individual strains before (rarely so at the root interface), to our knowledge it is the first report of a conserved regulation overlap within a complex microbial community.

Line228 "The results indicate that in planta activation of these processes is strongly linked with bacterial proliferation at roots and therefore suggests a possible link between the ability of diverse bacteria to utilize root-derived cues for their primary metabolism and their ability to dominate in the root microbiota". The link between activation of bacterial metabolic processes and bacterial proliferation is not demonstrated in this work. The ability of the root microbiota to utilize root-derived compounds have been amply demonstrated in previous works, as well as the bacterial ability to dominate in the community as a condition for root colonization.

We fully agree that we have not demonstrated a causal link here and we make it clearer in the text that those are correlations. We are absolutely aware of the important differences between correlation and causation and we are sorry if the sentences did not convey this clear message. We believe that the strength of this work is actually the last part in which we subject hypothesis to experimental testing and provide convincing evidence that our method is relevant to identify microbial genes that have physiological relevance for root colonization.

The author validated 3 genes using the bacterium *Rhodanobacter* sp. (Root179) as determinants of root colonization in mono-association assays. I was wondering if these genes are also relevant in the community context? This will emphasize the main conclusions of this manuscript.

We fully agree and we have now performed a new experiment in the FlowPot system using Root179 WT and mutant strains either inoculated alone or with the 105-member SynCom. We have designed and validated primer pairs that specifically amplify Root179 and not the other SynCom members (See validation in Supplementary Fig. 10b). These new results indicate that defect in root colonization observed in the agar-based matrix (mono-association) were retained in the FlowPot system (microbial community context) for Root179 Δ PstABCS and Δ typA mutants, indicating that this effect is very robust, irrespective of the system, the method or the presence of microbial competitors. However, this was not the case for the Δ ExbD mutant for which a defect in root colonization was observed in the agar-based matrix system but not in the FlowPot system. This indicates that the observed phenotype for this gene is system-dependent (See Fig. 6f,g). Note however that differences are expected given the major difference in growth systems (agar- vs peat-based), methods (CFU counts vs. qPCR) and biotic conditions (single strain vs. community context).

Do the authors believe that the peat matrix used could influence the final results of this work? Given that bacterial root colonization ability changes in response to environmental fluctuations, how robust are the validated gene functions under different abiotic conditions?

We have now validated the relevance of the genes in two fully independent systems (agar- and peat-based) and showed that defect in root colonization observed for Root179 Δ PstABCS and Δ typA mutants is very robust, irrespective of the system, the method or the presence of microbial competitors (See Fig. 6f,g and Supplementary Fig. 10b). However, we showed that for the Δ ExbD mutant, there is indeed a system-effect. Note that we actually used the agar-based system as an orthogonal assay to exactly address this question. We believe that the difference between the two systems that we used mimic very different abiotic conditions.

In general, I believe that this manuscript does not represent a step forward in the identification of new

bacterial genetic determinants of plant colonization and is not novel enough to be published in this journal.

To our knowledge, this is the first example showing the power of combining genome-resolved metatranscriptomics with SynCom reconstitution experiments in gnotobiotic system to simultaneously understand structural and functional architectures of multi-kingdom microbial communities colonizing plant roots at strain-level resolution. Our approach allowed us to go beyond descriptive work and to experimentally test whether microbial genes induced during proliferation at roots have physiological relevance for host colonization. Another unique aspect is that this strategy allowed us to look at conserved genes rather than lineage-specific innovations. There are numerous novel aspects here beyond the identification of bacterial genetic determinants (plant, bacteria, fungi multi-kingdom transcriptomes, strain-level resolution, analytical pipeline, community composition and comparison with amplicon sequencing, functional validation using targeted mutagenesis).

We thank the reviewer for the constructive feedback and hope that the new experiments and analyses that we now provide have clarified most of the concerns. We believe that the manuscript has greatly benefited from the constructive suggestions.

Reviewer #1 (Remarks to the Author):

The manuscript has been significantly improved. I have a few comments on newly added information during the revision.

The authors have added fungal transcriptome analysis as Supplementary Table 3, where they listed differentially expressed fungal genes between matrix and plant. I suggest presenting a big picture of the dataset as a figure, in addition to a supplementary table. A similar analysis presented in Figure 3 for fungi could be appropriate. Also, the authors could select representative fungal strains (perhaps those with high coverage) and show the transcriptome of each replicate in a heatmap and/or PCA plot. I believe a detailed presentation of data obtained with this novel approach is critical for publication.

For the analysis of Figure 4a, counting the number of strains differentially expressing each OG would be another reasonable option. The authors can first create a binary matrix of OG x Strain with DEG/non-DEG information, then count how many strains showed each OG as DEG. I am not certain if this approach is different from what the authors described as "only counting the number of strains showing differential OG enrichment," but I believe the analysis is complementary to the current analysis in Figure 4a and can support the conclusion.

L170: please remove "and"

Reviewer #2 (Remarks to the Author):

This is a very thorough and thoughtful review document accompanied by sufficient edits. My primary concerns about insufficient methods explanations have been addressed. I think the other edits and experiments in response to other reviewers also go beyond the minimum that would be required for publication.

Reviewer #3 (Remarks to the Author):

The authors of the manuscript have addressed all of my comments. I think the manuscript has improved significantly after revision. I want to thank all the authors for taking my comments so seriously.

Reviewer #1 (Remarks to the Author):

The manuscript has been significantly improved. I have a few comments on newly added information during the revision.

The authors have added fungal transcriptome analysis as Supplementary Table 3, where they listed differentially expressed fungal genes between matrix and plant. I suggest presenting a big picture of the dataset as a figure, in addition to a supplementary table. A similar analysis presented in Figure 3 for fungi could be appropriate.

In our supplementary data we provide a table describing for each fungal strain the genes regulation (DESeq2 differential expression). We believe this is sufficient to describe fungal transcriptomes regulation and we think that a new figure would not provide new information and would be fully redundant. In addition, we think that Figure 3 is useful to compare a large number of transcriptomes, but for fungi only 1-2 fungal strains had sufficient transcriptome coverage for such analysis.

Also, the authors could select representative fungal strains (perhaps those with high coverage) and show the transcriptome of each replicate in a heatmap and/or PCA plot. I believe a detailed presentation of data obtained with this novel approach is critical for publication.

We accordingly produced a PCOA of fungal transcriptomes based on OGs relative expression. We produced a PCOA with all the fungi for which enough reads were detected for the DESeq2 normalization (i.e. 11 fungi). The results highlight, like for bacteria, a differentiation between roots and matrix in the transcriptomic profiles, while the taxonomy also strongly determines the transcriptomic profile. These figures are now part of figure 3, (panels d and e).

For the analysis of Figure 4a, counting the number of strains differentially expressing each OG would be another reasonable option. The authors can first create a binary matrix of OG x Strain with DEG/non-DEG information, then count how many strains showed each OG as DEG. I am not certain if this approach is different from what the authors described as “only counting the number of strains showing differential OG enrichment,” but I believe the analysis is complementary to the current analysis in Figure 4a and can support the conclusion.

As mentioned in our rebuttal letter, doing this is also not a perfect strategy and has a major weakness (i.e. the log₂FC values will not be taken into account). We believe that providing two different ways of analyzing these data will confuse the reader rather than inform.

L170: please remove “and”

Done.

Reviewer #2 (Remarks to the Author):

This is a very thorough and thoughtful review document accompanied by sufficient edits. My

primary concerns about insufficient methods explanations have been addressed. I think the other edits and experiments in response to other reviewers also go beyond the minimum that would be required for publication.

We thank the reviewer for the comment acknowledging the work we did.

Reviewer #3 (Remarks to the Author):

The authors of the manuscript have addressed all of my comments. I think the manuscript has improved significantly after revision. I want to thank all the authors for taking my comments so seriously.

We thank the reviewer for the comment acknowledging the work we did.